# Enhancing In-Context Learning Performance with just SVD-Based Weight Pruning: A Theoretical Perspective

**Xinhao Yao[1], Xiaolin Hu[1], Shenzhi Yang[1], Yong Liu[1]**[*]
[1]Renmin University of China, Beijing, China
{yaoxinhao021978, xiaolinhu, yangshenzhi2003, liuyonggsai}@ruc.edu.cn

## Abstract

Pre-trained large language models (LLMs) based on Transformer have demonstrated striking in-context learning (ICL) abilities. With a few demonstration input-label pairs, they can predict the label for an unseen input without any parameter updates. In this paper, we show an exciting phenomenon that SVD-based weight pruning can enhance ICL performance, and more surprising, pruning weights in deep layers often results in more stable performance improvements than in shallow layers. However, the underlying mechanism of those findings still remains an open question. To reveal those findings, we conduct an in-depth theoretical analysis by presenting the implicit gradient descent (GD) trajectories of ICL and giving the mutual information based generalization bounds of ICL via full implicit GD trajectories. This helps us reasonably explain the surprising experimental findings. Besides, based on all our experimental and theoretical insights, we intuitively propose a simple, model-compression and derivative-free algorithm for downstream tasks in enhancing ICL inference. Experiments on benchmark datasets and open source LLMs display the method effectiveness[2].

## 1 Introduction

Recently, large language models (LLMs) based on the Transformer architecture [43] have emerged striking in-context learning (ICL) capabilities: Given a prompt containing demonstration sample and a test data, the model can make a prediction for the test data and achieve excellent performance without any parameter updates [7, 26, 34, 6]. This leads considerable works that aim to shed light on it [54, 21, 59, 3, 10, 16, 2, 46, 35] .

In this paper, we show our surprising findings in ICL inference by experimentally analyzing the effect of singular value decomposition (SVD)-based pruning on the model performance at different depth layers. As demonstrated in Figure 1: (i) SVD-based weight pruning can enhance ICL performance. Across all cases, it is evident that SVD-based weight pruning can generally enhance model performance at various depth layers, compared to the baseline performance without weight pruning (indicated by the dashed lines); (ii) Pruning weights in deep layers often results in more stable performance improvements than in shallow layers. Specially, deep layers weight matrices can be drastically reduced without much degradation in model performance, and may even get large improvements on it, while the model performance collapses after a sharp reduction at the shallow layers (see Section 2 for details). A similar case in Sharma et al. [38] notes that a large portion of singular values can be removed from linear layers in large language models without affecting or even improving reasoning performance. However, the underlying mechanism of this phenomenon still remains a mystery, and this paper seeks to explore the issue from the following two aspects.

---

[*]Corresponding author.
[2]The code is available at https://github.com/chen123CtrlS/EnhancingICL_SVDPruning.

38th Conference on Neural Information Processing Systems (NeurIPS 2024).

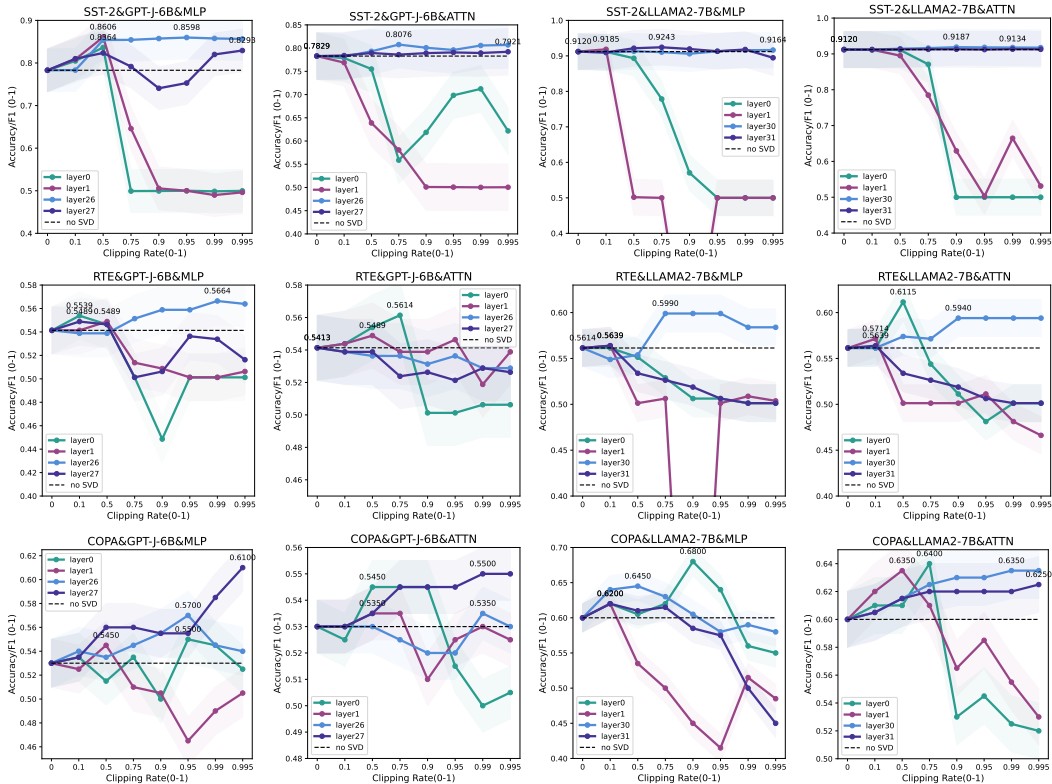

Figure 1: The effect of weight pruning across different layer types. The figure shows the phenomenon observed on the benchmark datasets (SST-2, RTE, COPA) and open source LLMs (GPT-J-6B and LLAMA2-7B). Each sub-figure corresponds only to the indicated type of dataset, model and module. Notice that this figure mainly focuses on exhibiting the impact of weight pruning to the first two and the last two layers of the model and different colors are used to distinguish between these layers. The dashed line represents the pretrained model performance without SVD. We operate on the whole of MLP or ATTN and specifically marked the points of highest performance. The amount of weight pruning is severe, for instance, the highest model performance sometimes occurs at a clipping rate of 0.995. This is about 99.5% of the matrix's original rank. For the definitions of "deep" and "shallow", please refer to Appendix B.2.

**Why this phenomenon?** We conduct an in-depth theoretical analysis to explain the findings. To be more specific, we first analyse the ICL form with Transformer from the perspective of ICL as implicit gradient descent fine-tuning, and we present the implicit gradient descent trajectories of ICL in **Theorem 1** in Section 3.2. Afterwards, we use the information-theoretic approaches [55, 32, 51] to give the generalization bounds of ICL via full implicit GD trajectories in **Theorem 2** in Section 3.3, explaining (**Q1**) why SVD-based weight pruning can enhance ICL performance? and (**Q2**) why do deep and shallow layers exhibit different behaviors when their weight matrices are drastically reduced?

**How to better use this phenomenon (Q3)?** Based on all our experimental and theoretical insights, we intuitively propose a simple, derivative-free and model-compression algorithm in Section 3.4 for downstream tasks in enhancing ICL inference, providing developers and researchers with an effective approach for handling downstream tasks. Experimental results for our algorithm on benchmark datasets [39, 58, 8, 12, 5, 11, 18] and open source LLMs [48, 42] verify that the method can visibly influence performance across different language understanding tasks in ICL inference.

Our primary objective is to establish a general theoretical framework that uncovers the underlying mechanism behind the phenomenon that SVD-based weight pruning can enhance ICL performance. Based on our theoretical insights, one can design new ICL algorithms. Accordingly, we did not directly compare our approach with other pruning methods. The algorithm in Section 3.4 is presented solely to illustrate how theoretical analysis can guide experimental procedures effectively.

## 1.1 Related Works

**Model compression.** In recent years, there has been growing theoretical and experimental evidence that models can be significantly pruned with very little drop in accuracy, thereby significantly reducing their storage requirements. To name a few, Frankle and Carbin [15] indicate that neural networks can typically have over 90% of their weights eliminated with little to no loss in performance. Analogous results have been demonstrated in both feed-forward and convolutional networks used for image classification [24, 49, 22, 37]. More specifically, Sharma et al. [38] present that careful pruning done at specific layers of Transformer models can produce significant boosts in performance on some tasks. The discovery of this phenomenon heightens interest in the connection between generalization and over-parameterization [56, 57], it also spurs research into developing pruning strategies that facilitate efficient model inference [30]. Additionally, the works [19, 40] find that performance remains nearly unchanged until a significant portion (up to half) of the layers are removed.

**In-context learning and gradient descent.** In order to better understand the ICL capabilities, considerable works try to understand ICL capabilities from the perspective of gradient descent. Irie et al. [21] and Dai et al. [10] explain ICL as implicit fine-tuning by figuring out a dual form of gradient descent for linear Transformer attention. However, the linear attention setting is less commonly used than Softmax attention in the LLMs and the details of gradient descent such as the choice of loss function and training data have not been clearly defined. Therefore, by using weight construction, von Oswald et al. [46] show the equivalence of linear self-attention mechanism and gradient descent on a linear regression task and Akyiurek et al. [3] prove that based on gradient descent and closed-form ridge regression, Transformers can learn linear models as learning algorithms. Further without using weight construction, Ren and Liu [35] connect Softmax attention with kernels and then give a novel interpretation that ICL with Transformer is really equivalent to a contrastive learning pattern.

**Information-theoretic generalization bounds.** The information-theoretic generalization bounds have been developed to analyze the expected generalization error of a learning algorithm. Given that they are dependent of distribution and algorithm, they are ideal tools to study the generalization behaviour of models performed with a specific algorithm. Russo and Zou [36] and Xu and Raginsky [55] first propose the Mutual information (MI) based bounds, which are then strengthened by additional techniques [4, 31, 20, 51]. Specifically, Negrea et al. [31] derive MI-based bounds by developing a PAC-Bayes-like bounding technique and Wang and Mao [51] develop generalization bounds for SGD via constructing an auxiliary iterative noisy process.

## 2 SVD-Based Weight Pruning can Enhance ICL

This section shows our surprising findings in ICL inference by performing a motivating analysis of three benchmark datasets in conjunction with two open source LLMs, the details are as follows. We choose GPT-J (6B,28 layers) [48] and LLAMA2 (7B,32 layers) [42] as our primary models for investigation, owing to their robust ICL performance and moderate model size, which align with our hardware resource. The attention (ATTN) layers are made up of key, query, value, out matrices both in GPT-J-6B and LLAMA2-7B. The mlp (MLP) layers in GPT-J-6B consist of input, output matrices, while the MLP layers in LLAMA2-7B are made up of up, gate, down matrices. For datasets, we use SST-2 [39] for sentiment analysis, RTE [5] for textual entailment and COPA [18] for causal inference. Details regarding datasets and how they were used are shown in Appendix C.1 and C.3.

Specially, we use the optimal rank-$r$ approximation mentioned later as SVD-based weight pruning method to show the effect of weight pruning across different layer and module types in Section 2.1, then we further analyze the effect of the ICL shot numbers on it in Section 2.2.

**The optimal rank-$r$ approximation and SVD.** Given a matrix $\mathbf{W} \in \mathbb{R}^{m \times n}$ and a constant $r \leq \min(m,n), r \in \mathbb{N}$. Eckart–Young–Mirsky theorem [14] provides an optimal solution $\mathbf{W}^*(r) = \mathbf{U}_{:r}\mathbf{\Sigma}_{:r}\mathbf{V}_{:r}^T$, $\mathbf{U}_{:r}, \mathbf{V}_{:r}$ are matrices containing the singular vectors corresponding to the largest $r$ singular values $[\sigma]_1^r$. Let $\xi = 1 - \frac{r}{\min(m,n)} \in (0,1)$ be the clipping rate.

### 2.1 Effect of Weight Pruning across Different Layer and Module Types

We plot the results of applying various amounts of clipping rate $\xi$ to each module matrices in the Transformer architecture on the corresponding evaluation index for each dataset, as depicted in Figure

1. These plots are grouped, such that each sub-figure corresponds only to the indicated type of dataset, model and module. Notice that this investigation mainly focuses on assessing the impact of weight pruning to the first two and the last two layers of the model to further clarify the impact of layer depth on the model's performance and different colors are used to distinguish between these layers.

All sub-figures clearly show an interesting phenomenon about these models in ICL inference: SVD-based weight pruning can enhance ICL performance in both shallow and deep layers across different module types. More surprising, deep layers weight matrices can be drastically reduced without much degradation in model performance, and may even get large improvements on it. This suggests that pruning weights in deep layers can effectively reduce model complexity while maintaining or enhancing model performance. And the model performance collapses to 0.5 (the expectation of a random guess in the binary task) after a sharp reduction at the shallow layers by contrast, indicating a higher sensitivity of the model to changes in shallow layers.

Based on the surprising findings in ICL inference mentioned above, three questions are certain to arise: (**Q1**) Why SVD-based weight pruning can enhance ICL performance? (**Q2**) Why do deep and shallow layers exhibit different behaviors when their weight matrices are drastically reduced? (**Q3**) How can we better use the phenomena about ICL in downstream tasks? We will address the first two questions theoretically in Section 3.3 and give a heuristic algorithm in Section 3.4 to answer the last.

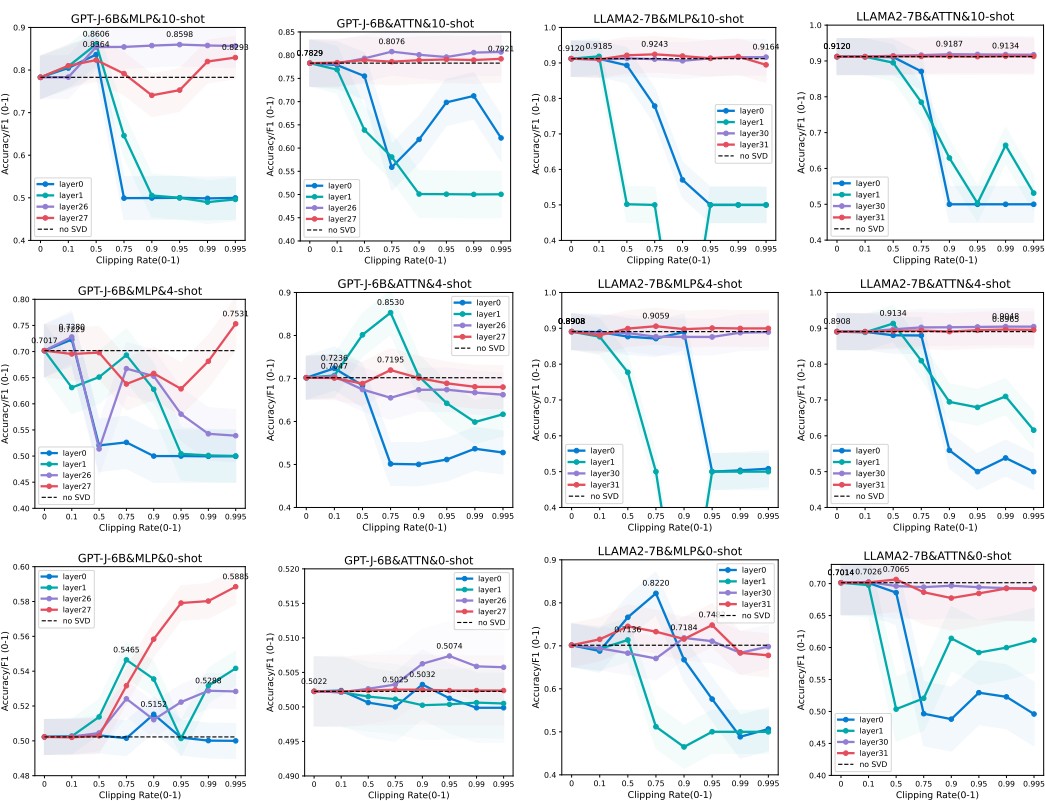

Figure 2: The effect of different ICL shot numbers is not uniform. Here we show the effect of different ICL shot numbers on the phenomenon mentioned in Section 2.1 as studied on the SST-2 dataset. Each row represents the results of the same shot numbers in different layers and modules, and each column represents the results of the different shot numbers in different layers of the same module. We also specifically marked the points of highest performance.

## 2.2 Effect of Different ICL Shot Numbers

Given that ICL can achieve higher performance with more demonstrations [23, 1], we further analyze the non-ignorable effect of different ICL shot numbers. To control for other influencing factors, we focus on SST-2 dataset [39] and retain the same test set for a single random seed as Section 2.1.

As shown in Figure 2, we compare the settings of three different ICL shot numbers: 0, 4 and 10. Following this, we analyze how the phenomenon changes across different shot numbers.

Firstly, we note that without weight pruning, the performance of the model improves with an increase in the number of ICL shots. Which is consistent with prior works. Besides, for every shot number, a phenomenon consistent with what is described in Section 2.1 is observed: SVD-based weight pruning can enhance ICL performance, pruning deep layer weight is more stable than pruning shallow weight. Last but not the least, Figure 2 also demonstrates roughly that with a decrease in the number of shots, the rate of performance collapse in the model slows down after a sharp reduction at the shallow layer. Intuitively, this is because LLMs exhibit a shift in focus in the ICL setting, which results in a reduced scope of the output space. The more shots there are, the more pronounced the shift becomes, and this also leads to a faster collapse. We also theoretically discuss it in Section 3.2 and 3.3 (**Remark 6**).

## 3    Theoretical Analysis Results

In this section, we first describe the core components of our study by reviewing some basic notations in Section 3.1, and present the implicit gradient descent trajectories of ICL in Section 3.2. Afterwards, we give a mutual information based generalization bounds of ICL via full implicit GD trajectories in Section 3.3. Based on all our experimental and theoretical insights, we intuitively propose a derivative-free and effective method for downstream tasks in enhancing ICL inference in Section 3.4. Complete proofs can be found in the Appendix. For ease of qualitative analysis, our theoretical analysis is mainly focuses on linear attention setting. We discuss the standard Softmax attention setting in Appendix A.2 and feed-forward (MLP) layers in Appendix A.3.

### 3.1    Preliminaries

We let $\mathcal{H}$ be the instance space and $\mu$ be an unknown distribution on $\mathcal{H}$, specifying random variable $\mathbf{h}$. In ICL setting, the model accepts a sequence of input $\mathbf{H} = [\mathbf{H}_s, \mathbf{h}_{N+1}]$ drawn i.i.d. from $\mu$, where $\mathbf{H}_s = [\mathbf{h}_1, \mathbf{h}_2, ..., \mathbf{h}_N]$ represents the demonstration sample and $\mathbf{h}_{N+1}$ is the test data. In the information-theoretic analysis framework, we let $\mathcal{W} \in \mathbb{R}^d$ be the space of hypotheses related to the model, and Transformer performs an implicit stochastic learning algorithm $\mathcal{A}$ which takes the demonstration sample $\mathbf{H}_s$ as its input and outputs a hypothesis $W \in \mathcal{W}$ according to some conditional distribution $Q_{W|\mathbf{H}_s}$. Similar to previous works [55, 51], we give the definition of expected generalization error.

**Expected generalization error.** Given a loss function $\ell : \mathcal{W} \times \mathcal{H} \to \mathbb{R}^+$, where $\ell(w, \mathbf{h})$ measures the "unfitness" or "error" of any $\mathbf{h} \in \mathcal{H}$ with respect to a hypothesis $w \in \mathcal{W}$. We take $\ell$ as a continuous function and assume that $\ell$ is differentiable almost everywhere with respect to $w$. The goal of learning is to find a hypothesis $w$ that minimizes the population risk, and for any $w \in \mathcal{W}$, the population risk is defined as $L_\mu(w) \triangleq \mathbb{E}_{\mathbf{h} \sim \mu}[\ell(w, \mathbf{h})]$. However, since $\mu$ via the sample $\mathbf{H}_s$ can only be partially observed, we instead turn to use the empirical risk, defined as $L_{\mathbf{H}_s}(w) \triangleq \frac{1}{N} \sum_{i=1}^N \ell(w, \mathbf{h}_i)$. Then the expected generalization error of $\mathcal{A}$ is defined as

$$\widetilde{\text{error}} \triangleq \mathbb{E}_{W, \mathbf{H}_s}[L_\mu(W) - L_{\mathbf{H}_s}(W)],$$

where the expectation is taken over $(\mathbf{H}_s, W) \sim \mu^N \otimes Q_{W|\mathbf{H}_s}$.

**In-context learning with Transformers[3].** By prompt design, most context semantic understanding tasks can be unified into classification tasks. Simplify the form of each token in $\mathbf{H}$ to $\mathbf{h}_i = [\mathbf{x}_i, \mathbf{y}_i]$, where $[\mathbf{x}_i]_1^N \in \mathbb{R}^{din}$ and $[\mathbf{y}_i]_1^N \in \mathbb{R}^{dout}$ are encoded input text and corresponding labels respectively. The test token has the form $\mathbf{h}_{N+1} = [\mathbf{x}_{N+1}, mask]$, where $mask$ is the label needed to predict, and usually set 0 as its initialization. Therefore, the form of attention with residual connection[4] is as follows:

$$\hat{\mathbf{H}} = \mathbf{H} + \mathbf{W}_V \mathbf{H} \mathbf{M} \, \text{Softmax} \left( \frac{((\mathbf{W}_K \mathbf{H})^T \mathbf{W}_Q \mathbf{H})}{\sqrt{d_{scale}}} \right), \tag{1}$$

---

[3]Please refer to Appendix B.1 for an explanation on how Eq.(1) and Eq.(2) can utilize the same mask.

[4]In real-world scenarios, the residual connection module in Transformers is indispensable.

where $\mathbf{W}_V, \mathbf{W}_K, \mathbf{W}_Q \in \mathbb{R}^{(dout+din) \times (dout+din)}$ are projection matrix, and the mask matrix $\mathbf{M} = \begin{pmatrix} \mathbf{I}_{N \times N} & 0 \\ 0 & 0 \end{pmatrix}$ is included in the attention. Specifically, $\hat{\mathbf{h}}_{N+1}$ can be formulated as

$$\hat{\mathbf{h}}_{N+1} = \mathbf{h}_{N+1} + \mathbf{W}_V \mathbf{H} \mathbf{M} \text{ Softmax} \left( \frac{((\mathbf{W}_K \mathbf{H})^T \mathbf{W}_Q \mathbf{h}_{N+1})}{\sqrt{d_{scale}}} \right), \tag{2}$$

and for ease of qualitative analysis, Eq.(2) can be approximated as a relaxed linear attention mechanism by removing the Softmax operation and scale factor:

$$\hat{\mathbf{h}}_{N+1} = \mathbf{h}_{N+1} + \mathbf{W}_V \mathbf{H} \mathbf{M} (\mathbf{W}_K \mathbf{H})^T \mathbf{W}_Q \mathbf{h}_{N+1}. \tag{3}$$

## 3.2 The Implicit Gradient Descent Trajectories of ICL

To begin with, we present the implicit gradient descent of ICL, inspired by [21, 10, 35]. These works describe how ICL with attention can be connected to a meta-optimizer which produces implicit gradient. The following lemma demonstrates the result.

**Lemma 1** (The Implicit Gradient Descent of ICL in a Single Linear Attention Layer). *Consider a Transformer consists of a single linear layer attention with residual connection, parameterized by $\mathbf{W}_V, \mathbf{W}_K, \mathbf{W}_Q$ as in Eq.(3). Same to Section 3.1, let $\mathbf{H} = [\mathbf{H}_s, \mathbf{h}_{N+1}]$ be the input, where $\mathbf{H}_s = [\mathbf{h}_1, \mathbf{h}_2, ..., \mathbf{h}_N]$ represents the demonstration sample and $\mathbf{h}_{N+1}$ is the test data. And let $\hat{\mathbf{h}}_{N+1}$ be the single layer output. Then, it holds that*

$$\Delta \mathbf{W}_{icl} = \mathbf{W}_V \mathbf{H}_s (\mathbf{W}_K \mathbf{H}_s)^T \mathbf{W}_Q = \left( \sum_{i=1}^{N} \mathbf{W}_V \mathbf{h}_i \otimes \mathbf{W}_K \mathbf{h}_i \right) \mathbf{W}_Q,$$

$$\hat{\mathbf{h}}_{N+1} = \mathbf{h}_{N+1} + \Delta \mathbf{W}_{icl} \mathbf{h}_{N+1},$$

*where $\mathbf{W}_V \mathbf{H}_s$ is regarded as the meta-gradient of ICL, which is used to generate the implicit gradient matrix $\Delta \mathbf{W}_{icl}$ to act on the final feature output. See Appendix A.1 for a proof.*

**Remark 1.** *It is worth noting that $rank(\Delta \mathbf{W}_{icl}) \leq rank(\mathbf{H}_s) \leq N$ by rank relations in matrix multiplication, indicating that this is a low-rank operation. This explains why the effect of different ICL shot numbers is not uniform in Section 2.2. Details of the discussion in standard Softmax attention setting can be found in Appendix A.2. Plus, a significant body of work has discovered that the ICL capabilities of models are not robust to the order of ICL sample. For instance, Lu et al. [28] observed that large language models (LLMs) are sensitive to the sequence of ICL examples, and Liu et al. [27] reported that the context-based ICL performance exhibits a U-shaped curve—models tend to perform better with information that appears at the beginning or at the end of the input context. This aligns with observations from actual model training where the sequence of training samples affects outcomes especially when training with a batch size of 1.*

According to the above, we further analyze the implicit gradient descent trajectories of ICL and provide the following Theorem. We will denote $\Delta \mathbf{W}_{icl}^t$ by $\Delta \mathbf{W}_t$ when there is no ambiguity, where $t$ represents $t$-th layer.

**Theorem 1** (The Implicit Gradient Descent Trajectories of ICL). *Consider a Transformer as a stack of $L$ linear attention blocks with residual connection, parameterized by $[\mathbf{W}_V^t]_1^L, [\mathbf{W}_K^t]_1^L, [\mathbf{W}_Q^t]_1^L$. Denote $[\mathbf{h}_i^t]_1^{N+1}$ as the output of the $t$-th layer, $[\mathbf{h}_i^0]_1^{N+1}$ as the initial input. Then for $t \in [L]$, it holds that*

$$\mathbf{G}_t = \Delta \mathbf{W}_t (1 + \mathbf{W}_{t-1}) = \Delta \mathbf{W}_t (1 + \mathbf{W}_0 + \sum_{j=1}^{t-1} \mathbf{G}_j),$$

$$\mathbf{h}_{N+1}^t = \mathbf{h}_{N+1}^0 + \sum_{j=1}^{t} \mathbf{G}_t \mathbf{h}_{N+1}^0 = \mathbf{h}_{N+1}^0 + \mathbf{W}_t \mathbf{h}_{N+1}^0,$$

*where $\Delta \mathbf{W}_t \triangleq \left( \sum_{i=1}^{N} \mathbf{W}_V^t \mathbf{h}_i^{t-1} \otimes \mathbf{W}_K^t \mathbf{h}_i^{t-1} \right) \mathbf{W}_Q^t$, $\mathbf{W}_0 = 0$, $\mathbf{W}_t = \mathbf{W}_{t-1} + \mathbf{G}_t$ and $\mathbf{G}_1 = \Delta \mathbf{W}_1$. See Appendix A.4 for a proof.*

**Remark 2.** *Note that the exclusion of Transformer weight ($[\mathbf{W}_K^t]_1^L, [\mathbf{W}_Q^t]_1^L, [\mathbf{W}_V^t]_1^L$) implies that $\mathbf{G}_t$ is only dependent on $\mathbf{W}_{t-1}$ and $\mathbf{H}_s$, this is consistent with gradient descent in terms of relevance.*

### 3.3 Generalization Bounds of ICL via Full Implicit GD Trajectories

In this part, for simplicity of representation, we flatten the weight matrix into a vector form $(\mathbf{vec}(\mathbf{W}_t) = W_t \in \mathbb{R}^d, \mathbf{vec}(\mathbf{G}_t) = G_t \in \mathbb{R}^d)$ and conduct our analysis within the weight and implicit gradient space of hypotheses $\mathcal{W}$ as detailed in Section 3.1. Notably, Ahn et al. [2] observe that, with the optimal parameters, a single layer linear Transformer implements a single step of preconditioned gradient descent and the preconditioning matrix not only adapts to the distribution of input data but also to the variance caused by data inadequacy. Garg et al. [16] show empirically that the trained Transformer is able to in-context learn the class of linear functions with respect to the prompt distribution, performing comparably to the optimal least squares estimator for any number of in-context examples considered, as exhibited in Figure 5a.

In addition, we also make a discussion on the noise of the implicit gradient in Section C.4, our analytical discussion indicates that the implicit gradients produced by Transformers in practical applications are noisy due to factors such as the extent of model pre-training and data characteristics (e.g., ICL shot number). We first present the assumption used in this subsection.

Let $G_t \triangleq \frac{1}{N} \sum_{i=1}^N \nabla \ell_i$ be the best implicit gradient of ICL that the model can produce, $\tilde{G}_t \triangleq \frac{1}{b} \sum_{i=1}^b \nabla \ell_i$ be the implicit gradient of ICL generated by the model in practical applications. $N$ is the threshold and $b$ is the the shot number in the actual input defined in Section C.4. And $V_t \triangleq G_t - \tilde{G}_t$ is the gradient noise caused by shot number, $C_t \triangleq \frac{N-b}{b(N-1)} (\frac{1}{N} \sum_{i=1}^N \nabla \ell_i \nabla \ell_i^T - G_t G_t^T)$ is the implicit gradient noise covariance. Similar to Wang and Mao [51]'s assumption in SGD, we approximate $V_t$ up to its second moment.

**Assumption 1.** *Assume the implicit gradient noise $V_t$ follows a Gaussian distribution, i.e., $V_t \sim \mathcal{N}(0, C_t)$, then in ICL implicit gradient descent trajectories,*

$$W_t = W_{t-1} - \eta \tilde{G}_t = W_{t-1} - \eta G_t + \eta C_t^{1/2} N_t, \tag{4}$$

*where $N_t \sim \mathcal{N}(0, I_d)$ is the standard Gaussian[5].*

**Remark 3.** *Regarding the validation of this assumption, empirical evidence from works [53, 25], suggests that SGD and Eq.(4) can achieve the similar testing performance. Additionally, we refer readers to some recent works [21, 16, 3, 10, 2, 46, 35], where the authors empirically verify that in-context learning of Transformer can achieve the similar testing performance to SGD. Together suggesting that studying Eq.(4) is arguably sufficient to understand generalization properties of ICL.*

**Theorem 1** indicates that the initial parameter $W_0 = 0$, which is independent of all other random variables. And an $L$-layer Transformer does implicit GD of ICL stops after $L$ updates, outputting $W_L$ as the implicit learned parameter. Our main results are mutual information based expected generalization error bounds of ICL, as presented in **Theorem 2**.

**Theorem 2** (The Generalization Bounds of ICL via Full Implicit Gradient Descent Trajectories)**.** *Under the conditions of **Theorem 1** and **Assumption 1**, assume the implicit gradient noise covariance $C_t$ is a positive-define matrix, the loss $\ell(w, \mathbf{h})$ is R-subGaussian for any $w \in \mathcal{W} \in \mathbb{R}^d$, then*

$$\widetilde{error} \leq \sqrt{\frac{R^2}{N} \sum_{t=1}^L \mathbb{E}_{\mathbf{W}_{t-1}}^{\mathbf{H}_s} \left[ d \log \left( \frac{\|\Delta \mathbf{W}_t\|_F^2 \cdot \left\| 1 + \sum_{j=1}^{t-1} \mathbf{G}_j \right\|_F^2 + \mathrm{tr}\{C_t\}}{d} \right) - \mathrm{tr}\{\log C_t\} \right]},$$

*where $\mathbf{vec}(\mathbf{G}_t) \in \mathbb{R}^d$, $\Delta \mathbf{W}_t \triangleq \left( \sum_{i=1}^N \mathbf{W}_V^t \mathbf{h}_i^{t-1} \otimes \mathbf{W}_K^t \mathbf{h}_i^{t-1} \right) \mathbf{W}_Q^t$ and $\mathbf{G}_t = \Delta \mathbf{W}_t (1 + \mathbf{W}_0 + \sum_{j=1}^{t-1} \mathbf{G}_j) = \Delta \mathbf{W}_t (1 + \mathbf{W}_{t-1})$. $\mathrm{tr}\{\cdot\}$ denotes the trace of a matrix, $\|\cdot\|_F$ denotes the Frobenius norm of a matrix and $\mathbb{E}_Y^X$ is the conditional expectation. Proof details in Appendix A.5.*

**Remark 4** (Deal with **Q1**)**.** *Theorem 2 indicates that one can control the generalization performance of ICL via controlling the implicit gradient norm along the entire ICL implicit GD trajectories. Specifically, modulating the norm of $[\mathbf{G}_t]_1^L$ or $[\Delta \mathbf{W}_t]_1^L$ may enhance performance when utilizing ICL. Note that controlling implicit gradient norm can also control the magnitude of the trace of implicit*

---

[5]Consider the continuous SDE: $dW = -\nabla L_{\mathbf{H}_s}(W)dt + [\eta C(W)]^{1/2} d\theta_t$, where $C(W)$ is the gradient noise covariance at $W$ and $\theta_t$ is a Wiener process. We can view Eq. (4) as discretization of the SDE.

*gradient noise covariance. This elucidates why weight pruning through SVD, even if it only alters a single weight matrix, can confer advantages on the performance of Transformers in ICL inference. We will present an example below demonstrating how weight pruning can affect the norm of $\mathbf{G}_t$ or $\Delta \mathbf{W}_t$, thereby influencing the expected generalization error. Additional examples provided in Appendix A.6. This is also why, as illustrated in Figure 1, the highest model performance sometimes occurs at a clipping rate of 0.995. Furthermore, this could elucidate the utility of normalization in Transformers. However, weight pruning also impacts the expressive power, implying that increased pruning does not invariably lead to better outcomes. This clarifies why the highest model performance may also occur at clipping rates lower than 0.995.*

**Example 1** (Prune $\mathbf{W}_Q, \mathbf{W}_K, \mathbf{W}_V$). *Consider* $\Delta \mathbf{W}_k = \left( \sum_{i=1}^N \mathbf{W}_V^k \mathbf{h}_i^{k-1} \otimes \mathbf{W}_K^k \mathbf{h}_i^{k-1} \right) \mathbf{W}_Q^k$ *when* $t = k$*. And we primarily consider the changes in a upper bound written as* $UB(\|\Delta \mathbf{W}_k\|_F^2) \triangleq \sum_{i=1}^N \left\| \mathbf{W}_V^k \mathbf{h}_i^{k-1} \otimes \mathbf{W}_K^k \mathbf{h}_i^{k-1} \right\|_F^2 \left\| \mathbf{W}_Q^k \right\|_F^2$*. Let $r$ represents the remained rank and $\delta$ represents the potential noise consisting of parts with small singular values, that is,* $\mathbf{W}_V^k = \mathbf{W}_{V_r}^k + \delta_V = \mathbf{U}_{:r}^V \mathbf{\Sigma}_{:r}^V (\mathbf{V}_{:r}^V)^T + \delta_V$*, the same operation is applied to $\mathbf{W}_Q^k$ and $\mathbf{W}_K^k$ as well. Then we have*

$$
\begin{aligned}
\left\| \mathbf{W}_V^k \mathbf{h}_i^{k-1} \right\|_2^2 &= \left\| (\mathbf{W}_{V_r}^k + \delta_V) \mathbf{h}_i^{k-1} \right\|_2^2 \\
&= \left\| \mathbf{W}_{V_r}^k \mathbf{h}_i^{k-1} \right\|_2^2 + 2(\mathbf{W}_{V_r}^k \mathbf{h}_i^{k-1})^T (\delta_V \mathbf{h}_i^{k-1}) + \left\| \delta_V \mathbf{h}_i^{k-1} \right\|_2^2 \\
&= \left\| \mathbf{W}_{V_r}^k \mathbf{h}_i^{k-1} \right\|_2^2 + \left\| \delta_V \mathbf{h}_i^{k-1} \right\|_2^2 \geq \left\| \mathbf{W}_{V_r}^k \mathbf{h}_i^{k-1} \right\|_2^2,
\end{aligned}
$$

*where* $\mathbf{W}_{V_r}^k = \mathbf{U}_{:r}^V \mathbf{\Sigma}_{:r}^V (\mathbf{V}_{:r}^V)^T$ *and* $\delta_V = \mathbf{U}_{r:}^V \mathbf{\Sigma}_{r:}^V (\mathbf{V}_{r:}^V)^T$*, and* $\mathbf{U}_{:r}, \mathbf{U}_{r:}$ *are orthometric (properties of SVD), and further have*

$$
\begin{aligned}
UB(\|\Delta \mathbf{W}_k\|_F^2) &\geq \sum_{i=1}^N \left\| \mathbf{W}_{V_r}^k \mathbf{h}_i^{k-1} \otimes \mathbf{W}_K^k \mathbf{h}_i^{k-1} \right\|_F^2 \left\| \mathbf{W}_Q^k \right\|_F^2 \qquad\qquad (*) \\
&\geq \sum_{i=1}^N \left\| \mathbf{W}_{V_r}^k \mathbf{h}_i^{k-1} \otimes \mathbf{W}_{K_r}^k \mathbf{h}_i^{k-1} \right\|_F^2 \left\| \mathbf{W}_Q^k \right\|_F^2 \\
&\geq \sum_{i=1}^N \left\| \mathbf{W}_{V_r}^k \mathbf{h}_i^{k-1} \otimes \mathbf{W}_{K_r}^k \mathbf{h}_i^{k-1} \right\|_F^2 \left\| \mathbf{W}_{Q_r}^k \right\|_F^2 = UB(\|\Delta \mathbf{W}_k(r)\|_F^2), \quad (**)
\end{aligned}
$$

*where Eq. (\*) is by*[6]

$$
\|\mathbf{vec}(\mathbf{A}) \otimes \mathbf{vec}(\mathbf{B})\|_F = \sqrt{\sum_i \sum_j |\mathbf{vec}(\mathbf{A})_i \mathbf{vec}(\mathbf{B})_j|^2} = \max_i |\sigma_i(\mathbf{vec}(\mathbf{A}) \otimes \mathbf{vec}(\mathbf{B}))|
$$

*and Eq. (\*\*) is by* $\|\mathbf{P}\|_F = \sqrt{\sum_i \sigma_i^2(\mathbf{P})}$ *for any matrix* $\mathbf{A}, \mathbf{B}$ *and* $\|\mathbf{P}\|_F \geq \|\mathbf{P}(r)\|_F$*.* $UB(\|\Delta \mathbf{W}_k(r)\|_F^2)$ *is the upper bound on* $\|\Delta \mathbf{W}_k\|_F^2$ *after using SVD.*

**Remark 5** (Deal with **Q2**). *It is notable that $\mathbf{G}_t$ is highly correlated with the sequence $(\Delta \mathbf{W}_1, ..., \Delta \mathbf{W}_t)$. More precisely, adjusting the weight matrix of the $k$-th layer, will invariably impact the norm of $[\Delta \mathbf{W}_t]_{t>k}^L$, further influence $[\mathbf{G}_t]_{t>k}^L$. When we calibrate the weight matrix of the $k$-th layer, the span of affected weight updates $\Delta \mathbf{W}_t$ encompasses $L - k + 1$ matrices, this indicates that the deeper the layer of the adjusted parameters is, the fewer the number of $\mathbf{G}_t$ affected. It also suggests that tweaks to the deep layers yield a more steadfast influence on the global norm of $[\mathbf{G}_t]_1^L$, thereby exerting a steadier influence over generalization performance. The enhanced stability observed in adjusting deeper layers, as demonstrated in our experimental findings in Section 2, can be theoretically explained based on the analysis presented above.*

**Remark 6** (How should **Theorem 2** be interpreted?). *Expected generalization error (**Theorem 2**) = population risk ($L_\mu$) - empirical risk ($L_{\mathbf{H}_s}$). More specifically, on the one hand, **Theorem 2** shows*

---

[6]For any vector $\mathbf{a}$ and $\mathbf{b}$, $\text{rank}(\mathbf{a} \otimes \mathbf{b}) = 1$, so the matrix $(\mathbf{a} \otimes \mathbf{b})$ only has one non-zero singular value. Combined with $\|\mathbf{P}\|_F = \sqrt{\sum_i \sigma_i^2(\mathbf{P})}$ and singular value is nonnegative, we can get $\|\mathbf{a} \otimes \mathbf{b}\|_F = \sqrt{\sum_i \sigma_i^2(\mathbf{a} \otimes \mathbf{b})} = \max_i[\sigma_i(\mathbf{a} \otimes \mathbf{b})]$, therefore, the unique non-zero singular value will decrease after performing SVD on $\mathbf{a}$ and/or $\mathbf{b}$.

*that clipping weights controls the F-norm of the implicit gradient ($[\Delta \mathbf{W}_t]_1^L/[\mathbf{G}_t]_1^L$), which helps reduce the expected generalization error. On the other hand, we can evaluate the empirical risk ($L_{\mathbf{H}_s}$) by assessing the model's performance on the validation set. If the generalization error is known, it is possible to estimate the population risk ($L_\mu$). Therefore, the most challenging aspect is addressing the generalization error. Furthermore, we will illustrate the interpretation of **Theorem 2** from the perspectives of pruning methods and the ICL shot numbers. (i) Adjusting the F-norm of ($[\Delta \mathbf{W}_t]_1^L/[\mathbf{G}_t]_1^L$) could enhance performance when utilizing ICL, suggesting that other weight-based pruning methods may also be effective. For example, magnitude-based pruning [52] directly controls the matrix F-norm. Certainly, there are also some layer-based pruning methods (e.g., drop-layer method [19]), we discuss in detail in Appendix B.6. (ii) Reducing the ICL shot numbers from $N$ to $N'$ can indeed lead to more robust outcomes specifically in the "SVD weight pruning" operation, rather than in the model's overall performance. Experimentally, in Section 2.2, we analyze the effect of different ICL shot numbers. As shown in Figure 2, with a decrease in the number of shots, the rate of performance collapse in the model slows down after a sharp reduction at the shallow layer. Theoretically, $\Delta \mathbf{W}(N) - \Delta \mathbf{W}(N') = \left( \sum_{i=N'+1}^{N} \mathbf{W}_V \mathbf{h}_i \otimes \mathbf{W}_K \mathbf{h}_i \right) \mathbf{W}_Q$, suggesting that the implicit gradient for $N$ is expected to be more sensitive than that for $N'$.*

### 3.4 Applications of Our Theoretical Understanding

In this portion, we deal with **Q3**. We further explore the relationship between model performance improvement and SVD, as illustrated by a simple case study.

**Case 1.** *Assuming that* $\mathbf{W}$ *has two unit eigenvectors* $\mathbf{x}_1, \mathbf{x}_2$, *based on the properties of eigenvectors:*

$$\mathbf{W}\mathbf{x_1} = \lambda_1 \mathbf{x_1}, \mathbf{W}\mathbf{x_2} = \lambda_2 \mathbf{x_2},$$

*and for a vector* $\mathbf{b}$ *on the hyperplane, it can be decomposed into a linear combination of two eigenvectors:*

$$\mathbf{b} = l_1 \mathbf{x_1} + l_2 \mathbf{x_2} = \mathbf{W} \left( \frac{l_1}{\lambda_1} \mathbf{x}_1 + \frac{l_2}{\lambda_2} \mathbf{x}_2 \right) = \mathbf{W}\mathbf{x},$$

*where* $l_1, l_2$ *are coefficients and* $\lambda_1, \lambda_2$ *are eigenvalues. Notice that if* $\lambda_1 \gg \lambda_2$, *when point* $\mathbf{b}$ *moves in the direction of* $\mathbf{x_1}$, *the value of* $l_1$ *changes but the solution set* $\mathbf{x}$ *changes insignificantly. Conversely, if it moves in the direction of* $\mathbf{x_2}$, *the value of* $l_2$ *changes and the solution set* $\mathbf{x}$ *changes dramatically, this is an ill-conditioned[7] problem whose solution is highly biased under small perturbations.*

Given that SVD can be considered to play a role in noise reduction in ICL inference, and that this noise can cause significant disturbances to model output. We thus introduce the matrix ill-conditionedness, which can be measured by the Matrix condition number defined as follows.

**Matrix condition number[8].** $Cond(\mathbf{W}) = ||\mathbf{W}||_p ||\mathbf{W}^{-1}||_p$. Specifically, the condition number $Cond(\mathbf{W}) = \frac{\sigma_{max}}{\sigma_{min}}$ when $p = 2$, where $\sigma_{max}, \sigma_{min}$ denote the maximum and minimum singular values respectively. Generally, for matrices of the same order, the higher the condition number of a matrix is, the greater its ill-conditionedness will be.

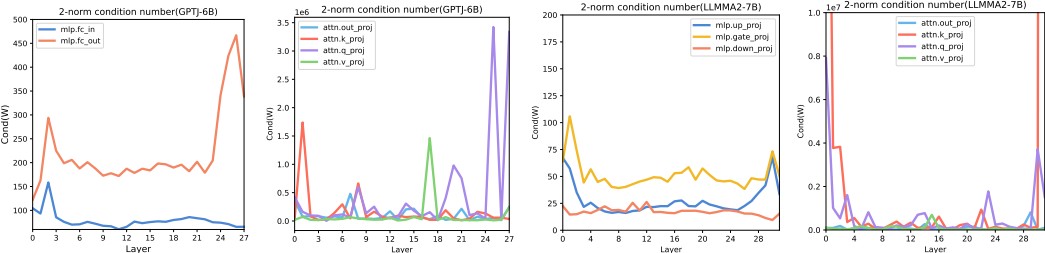

Figure 3: 2-norm condition number of GPTJ-6B&LLAMA2-7B. The condition numbers for MLP are significantly lower than those for ATTN. In deeper layers, condition numbers tend to be higher. These matrices are ill-conditioned for they satisfy $\sigma_{max} \gg \sigma_{min}$.

---

[7]Now consider a linear system $\mathbf{W}\mathbf{x} = \mathbf{b}$, where the solution set $\mathbf{x}$ is highly sensitive to the coefficients of $\mathbf{W}$ and $\mathbf{b}$. In such cases, the system of equations is termed ill-conditioned.

[8]Matrix condition number is an option, any indicator that can guide the control of norms is a potential option.

Then we analyze the matrix condition number of the models as shown in Figure 3. We observe that the condition number of deeper layers (especially the last layers) in the model is generally higher, indicating that the condition number may serve as a reference for adjusting the model.

Based on all our exciting insights, we find it intuitive to design ICL improvements based on them, especially in downstream tasks. We propose a method where we first select layers with the top-k largest condition numbers and then identify the layer with the largest number among these. We perform a greedy search for the optimal clipping rate $\xi^*$ on the validation set and subsequently evaluate it on the test set. This procedure is reported in **Algorithm 1** in Appendix C.2.

**Experimental setup.** We conduct experiments for **Algorithm 1** on widely adopted benchmark datasets, including SST-2 [39], AGNEWS [58], EMOC [8], MRPC [12], RTE [5], CB [11], COPA [18]. Please refer to Appendix C.1 and C.3 for more detailed dataset and prompt settings. And for models, we use the same models as in Section 2.

**Experimental results.** As Figure 4 shows, the results indicate that **Algorithm 1** can visibly influence performance across different language understanding tasks in ICL inference. Specially, the effectiveness of **Algorithm 1** in enhancing performance varies not only by the model but also significantly by the task, indicating a potential task-specific and model-specific threshold for the benefits derived from algorithmic enhancements. More notably, **Algorithm 1** is a derivative-free optimization method and compresses the model to some extent, providing developers and researchers with an effective approach for handling downstream tasks. We also invite readers to refer to Appendices B.3, B.4 and B.5 for discussions on *What would be the effect of pruning only a single module?*, *Why optimal clipping rate $\xi$ varies?* and *What would happen if we apply the same clipping rate to other datasets*?

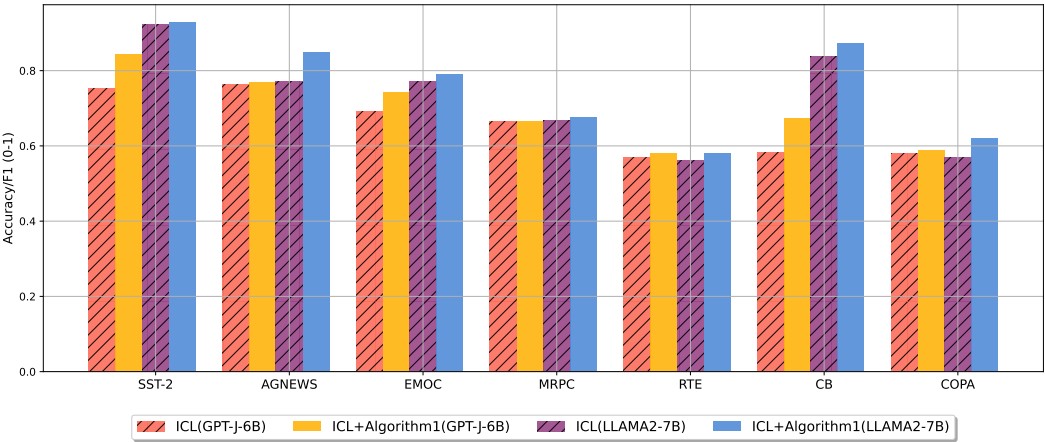

Figure 4: The Model Performance on Test set by different tasks. The results are obtained by comparing four scenarios: ICL (GPT-J-6B), ICL+Algorithm1 (GPT-J-6B), ICL (LLAMA2-7B) and ICL+Algorithm1 (LLAMA2-7B). ICL+Algorithm1 demonstrates superior results over only ICL on different tasks. See Appendix C.5 for detailed numbers.

## 4 Conclusion and Limitation

In this paper, we show our surprising findings in ICL inference: SVD-based weight pruning can enhance ICL performance both in shallow and deep layers across different module types, and pruning weights in deep layers often results in more stable performance improvements than in shallow layers. We conduct an in-depth theoretical analysis and explanation of these findings. Specifically, we first present the implicit gradient descent trajectories of ICL, afterwards, we give a mutual information based generalization bounds of ICL via full implicit GD trajectories. Based on all our experimental and theoretical insights, we intuitively propose a derivative-free and effective method for downstream tasks in enhancing ICL inference. However, further studies are required on (i) how to extend our generalization theory to a more standard Transformer architecture, (ii) do the results about ICL hold true for tasks beyond natural language processing and (iii) how to minimize the cost of searching the optimal clipping rate. Those will deepen our understanding of the ICL capabilities.

## Acknowledgments

This research was supported by National Natural Science Foundation of China (No.62476277, No.6207623), Beijing Natural Science Foundation (No.4222029), CCF-ALIMAMA TECH Kangaroo Fund(No.CCF-ALIMAMA OF 2024008), and Huawei-Renmin University joint program on Information Retrieval. We also acknowledge the support provided by the fund for building worldclass universities (disciplines) of Renmin University of China and by the funds from Beijing Key Laboratory of Big Data Management and Analysis Methods, Gaoling School of Artificial Intelligence, Renmin University of China, from Engineering Research Center of Next-Generation Intelligent Search and Recommendation, Ministry of Education, from Intelligent Social Governance Interdisciplinary Platform, Major Innovation & Planning Interdisciplinary Platform for the "DoubleFirst Class" Initiative, Renmin University of China, from Public Policy and Decision-making Research Lab of Renmin University of China, and from Public Computing Cloud, Renmin University of China.

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

# A Omitted Proofs and Additional Results

## A.1 Proof of Lemma 1

*Proof.* Given that numerous studies [21, 10, 35] describe ICL with attention can be connected to a meta-optimizer which produces implicit gradient, we show the dual form between linear attention and gradient descent. First, consider a very simple linear layer,

$$F(\mathbf{x}) = \mathbf{W}\mathbf{x},$$

where $\mathbf{W} \in \mathbb{R}^{dout \times din}$ is the projection matrix. Given training inputs $[\mathbf{x}_i]_{i=1}^N \in \mathbb{R}^{din}$ with their labels $[\mathbf{y}_i]_{i=1}^N \in \mathbb{R}^{dout}$ and the loss function $\mathcal{L}$ with learning rate $\eta$, gradient descent process produces the corresponding back-propagation signals $[\mathbf{e}_i]_{i=1}^N \in \mathbb{R}^{dout}$, where $\mathbf{e}_i = -\eta \left( \nabla_{\mathbf{y}_i'} \mathcal{L} \right)$ and $\mathbf{y}_i' = \mathbf{W}\mathbf{x}_i$. During test time, we can use a trained linear layer,

$$\hat{F}(\mathbf{x}_{test}) = \hat{\mathbf{W}}\mathbf{x}_{test} = (\mathbf{W} + \Delta\mathbf{W})\mathbf{x}_{test} = \mathbf{W}\mathbf{x}_{test} + \left( \sum_{i=1}^N \mathbf{e}_i \otimes \mathbf{x}_i \right) \mathbf{x}_{test}, \qquad (5)$$

where $\otimes$ denotes the outer product according to the chain rule of differentiation. On the other hand, this process can be associated with linear attention,

$$LinearAttn(\mathbf{V}, \mathbf{K}, \mathbf{q}) = \mathbf{V}\mathbf{K}^T\mathbf{q} = \sum_{i=1}^N \mathbf{v}_i(\mathbf{k}_i^T\mathbf{q}) = \sum_{i=1}^N (\mathbf{v}_i \otimes \mathbf{k}_i)\mathbf{q}, \qquad (6)$$

where $[\mathbf{k}_i]_1^N, [\mathbf{v}_i]_1^N \in \mathbb{R}^{din}$ represent the key and value vectors respectively, forming the key and value matrix $\mathbf{K}, \mathbf{V} \in \mathbb{R}^{din \times N}$ in the attention mechanism. Based on Eq.(6), rewrite Eq.(5) as:

$$\hat{F}(\mathbf{x}_{test}) = \mathbf{W}\mathbf{x}_{test} + \left( \sum_{i=1}^N \mathbf{e}_i \otimes \mathbf{x}_i \right) \mathbf{x}_{test} = \mathbf{W}\mathbf{x}_{test} + LinearAttn(\mathbf{E}, \mathbf{X}, \mathbf{x}_{test}). \qquad (7)$$

Then let's go back to the ICL process and approximate standard attention Eq.(2) as a relaxed linear attention mechanism by removing the Softmax operation and scale factor:

$$\begin{aligned}
F_{icl}(\mathbf{h}_{N+1}) \approx \hat{\mathbf{h}}_{N+1} &= \mathbf{h}_{N+1} + \mathbf{W}_V\mathbf{H}\mathbf{M}(\mathbf{W}_K\mathbf{H})^T\mathbf{W}_Q\mathbf{h}_{N+1} \\
&= \mathbf{h}_{N+1} + \mathbf{W}_V\mathbf{H}_s(\mathbf{W}_K\mathbf{H}_s)^T\mathbf{W}_Q\mathbf{h}_{N+1} \\
&= \mathbf{h}_{N+1} + \mathbf{W}_0\mathbf{h}_{N+1} + \mathbf{W}_V\mathbf{H}_s(\mathbf{W}_K\mathbf{H}_s)^T\mathbf{W}_Q\mathbf{h}_{N+1} \\
&= \mathbf{h}_{N+1} + \mathbf{W}_0\mathbf{h}_{N+1} + \left( \sum_{i=1}^N \mathbf{W}_V\mathbf{h}_i \otimes \mathbf{W}_K\mathbf{h}_i \right) \mathbf{W}_Q\mathbf{h}_{N+1} \\
&= \mathbf{h}_{N+1} + \mathbf{W}_0\mathbf{h}_{N+1} + LinearAttn(\mathbf{W}_V\mathbf{H}_s, \mathbf{W}_K\mathbf{H}_s, \mathbf{W}_Q\mathbf{h}_{N+1}) \\
&= \mathbf{h}_{N+1} + \mathbf{W}_0\mathbf{h}_{N+1} + \Delta\mathbf{W}_{icl}\mathbf{h}_{N+1}
\end{aligned} \qquad (8)$$

Where $\mathbf{W}_0 = 0$, and $\mathbf{W}_V\mathbf{H}_s$ is regarded as the meta-gradient of ICL, which is used to generate the implicit gradient matrix $\Delta\mathbf{W}_{icl}$ to act on the final feature output. $\square$

## A.2 Softmax Attention and ICL Implicit GD

Inspired by Ren and Liu [35], by connecting Softmax Attention with Kernels, we can interpret ICL as a gradient descent process in contrast learning pattern. For simplicity, as in Section 3.1, scale factors are removed as follows:

$$\begin{aligned}
\mathbf{A} = \text{Softmax}((\mathbf{W}_K\mathbf{H})^T\mathbf{W}_Q\mathbf{H}), \mathbf{A}_u = \exp((\mathbf{W}_K\mathbf{H})^T\mathbf{W}_Q\mathbf{H}), \\
\mathbf{A} = \mathbf{D}^{-1}\mathbf{A}_u, \mathbf{D} = diag(\mathbf{A}_u, \mathbf{1}_N).
\end{aligned} \qquad (9)$$

where $\exp(\cdot)$ is element-wise. Following Choromanski et al. [9] and Ren and Liu [35], a Softmax kernel function $\mathbf{K}_{\text{Softmax}} : \mathbb{R}^{dout} \times \mathbb{R}^{dout} \to \mathbb{R}_+$ is defined as:

$$\mathbf{K}_{\text{Softmax}}(\mathbf{x}, \mathbf{y}) = \exp(\mathbf{x}^T\mathbf{y}) = \exp\left( \frac{\|\mathbf{x}\|^2}{2} \right) \mathbf{K}_{gauss}(\mathbf{x}, \mathbf{y}) \exp\left( \frac{\|\mathbf{y}\|^2}{2} \right), \qquad (10)$$

where $\mathbf{K}_{gauss}(\mathbf{x}, \mathbf{y}) = \exp\left(-\frac{\|\mathbf{x}-\mathbf{y}\|^2}{2}\right)$ is the Gaussian kernel with a variance $\sigma^2 = 1$. According to Mercer's theorem [29], there exists some mapping function $\phi$ satisfying $K_{\text{Softmax}}(\mathbf{x}, \mathbf{y}) = \phi(\mathbf{x})^T \phi(\mathbf{y})$. Thus, rewrite Eq.(9) as:

$$
\begin{aligned}
\mathbf{A}_u(i, j) &= \exp((\mathbf{W}_K \mathbf{h}_i)^T \mathbf{W}_Q \mathbf{h}_j) \\
&= \mathbf{K}_{\text{Softmax}}(\mathbf{W}_K \mathbf{h}_i, \mathbf{W}_Q \mathbf{h}_j) \\
&= \phi(\mathbf{W}_K \mathbf{h}_i)^T \phi(\mathbf{W}_Q \mathbf{h}_j),
\end{aligned}
\tag{11}
$$

then, connect with Eq.(2) we have,

$$
\begin{aligned}
\hat{\mathbf{h}}_{N+1} &= \mathbf{h}_{N+1} + \mathbf{W}_V \mathbf{HM} \text{ Softmax} \left( \frac{((\mathbf{W}_K \mathbf{H})^T \mathbf{W}_Q \mathbf{h}_{N+1})}{\sqrt{d_{scale}}} \right) \\
&\approx \mathbf{h}_{N+1} + \mathbf{W}_V \mathbf{HM} \text{ Softmax} \left( ((\mathbf{W}_K \mathbf{H})^T \mathbf{W}_Q \mathbf{h}_{N+1}) \right) \\
&= \mathbf{h}_{N+1} + \frac{1}{D'} \mathbf{W}_V [\mathbf{H}_s, \mathbf{h}_{N+1}] \mathbf{M} [\phi(\mathbf{W}_K \mathbf{H}_s), \phi(\mathbf{W}_K \mathbf{h}_{N+1})]^T \phi(\mathbf{W}_Q \mathbf{h}_{N+1}) \\
&= \mathbf{h}_{N+1} + \frac{1}{D'} \mathbf{W}_V \mathbf{H}_s \phi(\mathbf{W}_K \mathbf{H}_s)^T \phi(\mathbf{W}_Q \mathbf{h}_{N+1}) \\
&= \mathbf{h}_{N+1} + \frac{1}{D'} \left[ \sum_{i=1}^N (\mathbf{W}_V \mathbf{H}_s)_i \otimes \phi(\mathbf{W}_K \mathbf{H}_s)_i \right] \phi(\mathbf{W}_Q \mathbf{h}_{N+1}) \\
&= \mathbf{h}_{N+1} + \Delta \mathbf{W}'_{icl} \phi(\mathbf{W}_Q \mathbf{h}_{N+1}),
\end{aligned}
\tag{12}
$$

where $D' = \mathbf{1}_N \phi(\mathbf{W}_K \mathbf{H}_s)^T \phi(\mathbf{W}_Q \mathbf{h}_{N+1}) + \phi(\mathbf{W}_K \mathbf{h}_{N+1})^T \phi(\mathbf{W}_Q \mathbf{h}_{N+1})$ represents a constant for the normalized attention scores. It can be considered that the mapping function $\phi$, or rather the effect of the Softmax function, is to project the original features into a higher-dimensional space to capture more profound features. Subsequently, learning and meta-optimization are conducted under this new feature space. It is also noteworthy that $\text{rank}(\Delta \mathbf{W}'_{icl}) \leq \text{rank}(\mathbf{H}_s) \leq N$.

### A.3 Feed-Forward and ICL Implicit GD

The LLMs based on the Transformer architecture not only contain an Attention mechanism, but also include feed-forward (MLP) layers. It can be estimated that the feed-forward layers account for approximately two-thirds of the parameters in a Transformer model [45, 44]. Thus, MLP layers are essential for LLMs and there has been considerable work hypothesizing and analyzing the model's feed-forward layers. For example, Geva et al. [17], Qiu et al. [33] show that feed-forward layers in Transformer based language models operate as key-value memories, Tian et al. [41] propose Joint MLP/Attention (JoMA) dynamics by integrating out the self-attention layers in Transformers, analyzes joint training of MLP and self-attention layers, and qualitatively explains dynamics of multi-layer Transformers. Based on the inspiration above, we take MLP into consideration and rewrite Eq. (3) as:

$$
\begin{aligned}
\hat{\mathbf{h}}_{N+1} &= \mathbf{h}_{N+1} + \text{MLP}(\mathbf{W}_V \mathbf{HM}(\mathbf{W}_K \mathbf{H})^T \mathbf{W}_Q \mathbf{h}_{N+1}) \\
&= \mathbf{h}_{N+1} + \mathbf{W}_{out} \text{relu}(\mathbf{W}_{in} \mathbf{W}_V \mathbf{HM}(\mathbf{W}_K \mathbf{H})^T \mathbf{W}_Q \mathbf{h}_{N+1}),
\end{aligned}
$$

where $\mathbf{W}_V, \mathbf{W}_K, \mathbf{W}_Q \in \mathbb{R}^{(dim1) \times (dout+din)}$, $\mathbf{W}_{in} \in \mathbb{R}^{(dim2) \times (dim1)}$, $\mathbf{W}_{out} \in \mathbb{R}^{(dout+din) \times (dim2)}$ are projection matrices and $\text{relu}(\cdot)$ is the activation function. And we can further relax the activation function for ease of qualitative analysis:

$$
\begin{aligned}
\hat{\mathbf{h}}_{N+1} &= \mathbf{h}_{N+1} + \mathbf{W}_{out} \mathbf{W}_{in} \mathbf{W}_V \mathbf{HM}(\mathbf{W}_K \mathbf{H})^T \mathbf{W}_Q \mathbf{h}_{N+1} \\
&= \mathbf{h}_{N+1} + \mathbf{W}_{MLP} \mathbf{W}_V \mathbf{HM}(\mathbf{W}_K \mathbf{H})^T \mathbf{W}_Q \mathbf{h}_{N+1},
\end{aligned}
$$

where $\mathbf{W}_{MLP} \in \mathbb{R}^{(dout+din) \times (dim1)}$, which can be seen as a dimensional adaptation. And then do the similar derivation that we do in Appendix A.1, we can get $\Delta \mathbf{W}''_{icl} = \mathbf{W}_{MLP} \left( \sum_{i=1}^N \mathbf{W}_V \mathbf{h}_i \otimes \mathbf{W}_K \mathbf{h}_i \right) \mathbf{W}_Q$.

## A.4 Proof of Theorem 1

*Proof.* Similar to A.1, Pay attention to the test token when $i = N + 1$,

$$\mathbf{h}_{N+1}^1 = \mathbf{h}_{N+1}^0 + \Delta\mathbf{W}_1\mathbf{h}_{N+1}^0,$$
$$\mathbf{h}_{N+1}^2 = \mathbf{h}_{N+1}^1 + \Delta\mathbf{W}_2\mathbf{h}_{N+1}^1$$
$$= \mathbf{h}_{N+1}^0 + \Delta\mathbf{W}_1\mathbf{h}_{N+1}^0 + \Delta\mathbf{W}_2(1 + \Delta\mathbf{W}_1)\mathbf{h}_{N+1}^0,$$
$$\mathbf{h}_{N+1}^3 = \mathbf{h}_{N+1}^2 + \Delta\mathbf{W}_3\mathbf{h}_{N+1}^2$$
$$= \mathbf{h}_{N+1}^0 + \Delta\mathbf{W}_1\mathbf{h}_{N+1}^0 + \Delta\mathbf{W}_2(1 + \Delta\mathbf{W}_1)\mathbf{h}_{N+1}^0$$
$$+ \Delta\mathbf{W}_3\left(1 + \Delta\mathbf{W}_1 + \Delta\mathbf{W}_2(1 + \Delta\mathbf{W}_1)\right)\mathbf{h}_{N+1}^0,$$
$$......$$

Thus, we give the implicit GD trajectories $[\mathbf{G}_t]_1^L$ of ICL, where $\mathbf{W}_t = \mathbf{W}_{t-1} + \mathbf{G}_t$,

$$\mathbf{W}_0 = 0,$$
$$\mathbf{G}_1 = \Delta\mathbf{W}_1,$$
$$\mathbf{G}_2 = \Delta\mathbf{W}_2(1 + \Delta\mathbf{W}_1) = \Delta\mathbf{W}_2(1 + \mathbf{G}_1),$$
$$\mathbf{G}_3 = \Delta\mathbf{W}_3\left(1 + \Delta\mathbf{W}_1 + \Delta\mathbf{W}_2(1 + \Delta\mathbf{W}_1)\right) = \Delta\mathbf{W}_3(1 + \mathbf{G}_1 + \mathbf{G}_2), \quad (13)$$
$$......$$
$$\mathbf{G}_L = \Delta\mathbf{W}_L(1 + \sum_{t=1}^{L-1}\mathbf{G}_t).$$

On the one hand, Eq.(13) shows $\mathbf{G}_t = \Delta\mathbf{W}_t(1 + \mathbf{W}_{t-1})$, on the other hand, $\Delta\mathbf{W}_t \triangleq \left(\sum_{i=1}^N \mathbf{W}_V^t\mathbf{h}_i^{t-1} \otimes \mathbf{W}_K^t\mathbf{h}_i^{t-1}\right)\mathbf{W}_Q^t$, which is only depend on $([\mathbf{h}_i^0]_1^N, [\mathbf{W}_K^t]_1^L, [\mathbf{W}_Q^t]_1^L, [\mathbf{W}_V^t]_1^L)$. So the exclusion of Transformer weight $([\mathbf{W}_K^t]_1^L, [\mathbf{W}_Q^t]_1^L, [\mathbf{W}_V^t]_1^L)$ implies that $\mathbf{G}_t$ is only dependent on $\mathbf{W}_{t-1}$ and $\mathbf{H}_s$. $\qquad\square$

And the prediction can be read from the corresponding position in $L$-th layer output $h_{N+1}^L$ as follows,

$$\mathbf{h}_{N+1}^L = \begin{pmatrix} * \\ \mathbf{y}_{pred} \end{pmatrix} = \mathbf{h}_{N+1}^0 + \mathbf{W}_L\mathbf{h}_{N+1}^0 = (1 + \mathbf{W}_L)\begin{pmatrix} \mathbf{x}_{N+1} \\ mask \end{pmatrix}.$$

## A.5 Proof of Theorem 2

The origin form of the mutual information based bound is predicated on a sample-specific MI, which quantifies the shared information between the output variable $W$ and the input sample set $\mathbf{H}_s$. The following lemma shows the result:

**Lemma 2.** *(Xu and Raginsky [55, Theorem 1.]). Assume the loss $\ell(w, \mathbf{h})$ is R-subGaussian for any $w \in \mathcal{W}$, then*

$$\widetilde{error} \leq \sqrt{\frac{2R^2}{N}I(W; \mathbf{H}_S)},$$

*where $I(W; \mathbf{H}_S) = D_{KL}(Q_{W,\mathbf{H}_S}\|Q_W \otimes Q_{\mathbf{H}_S})$ is the mutual information and $D_{KL}$ denotes the KL divergence.*

Unroll the terminal parameters' mutual information $I(W; \mathbf{H}_S)$ to the full trajectories' mutual information will get:

**Lemma 3.** *(Wang and Mao [51, Lemma 4.]). Let **Assumption 1** hold, then $I(W_L; \mathbf{H}_S) \leq \sum_{t=1}^L I(-G_t + C_t^{1/2}N_t; \mathbf{H}_S|W_{t-1})$. Let $-G_t + C_t^{1/2}N_t \triangleq \mathcal{G}_t$.*

*Proof.* Recall **Assumption 1** Eq.(4), Wang and Mao [51] get,

$$I(W_L; \mathbf{H}_S) = I(W_{L-1} + \eta(-G_L + C_L^{1/2}N_L); \mathbf{H}_S)$$

$$\leq I(W_L, \eta(-G_L + C_L^{1/2}N_L); \mathbf{H}_S) \tag{14}$$

$$= I(W_{L-1}; \mathbf{H}_S) + I(\eta(-G_L + C_L^{1/2}N_L); \mathbf{H}_S|W_{L-1}) \tag{15}$$

$$= I(W_{L-1}; \mathbf{H}_S) + I(-G_L + C_L^{1/2}N_L; \mathbf{H}_S|W_{L-1})$$

$$\leq \sum_{t=1}^{L} I(-G_t + C_t^{1/2}N_t; \mathbf{H}_S|W_{t-1}) + I(W_0; \mathbf{H}_S)$$

$$= \sum_{t=1}^{L} I(-G_t + C_t^{1/2}N_t; \mathbf{H}_S|W_{t-1}).$$

where Eq. (14) is by the data processing inequality (e.g., $Z - (X, Y) - (X + Y)$ form a Markov chain then $I(X + Y, Z) \leq I(X, Y; Z)$), Eq. (15) is by the chain rule of the mutual information, and learning rate $\eta$ is dropped since mutual information is scale-invariant. $I(W_0; \mathbf{H}_S) = 0$ for $W_0$ is independent of all other random variables in **Theorem 1** and **Assumption 1**. $\qquad\square$

Besides, we present the variational representation of mutual information:

**Lemma 4.** *(Polyanskiy and Wu [32, Corollary 3.1.]). For two random variables $X$ and $Y$, we have $I(X; Y) = \inf_P \mathbb{E}_X[D_{KL}(Q_{Y|X}\|P)]$, where the infimum is achieved at $P = Q_Y$.*

**Lemma 5.** *(Wang and Mao [51, Lemma 5.]). At every time step $t$, let $P_{\tilde{N}_t|W_{t-1}}$ be any distribution satisfying $D_{KL}(P_{\mathcal{G}_t|W_{t-1}}\|P_{\tilde{N}_t|W_{t-1}}) < \infty$, we have $I(\mathcal{G}_t; \mathbf{H}_s|W_{t-1}) = \mathbb{E}_{W_{t-1}}\left[\inf_{P_{\tilde{N}_t|W_{t-1}}} \mathbb{E}_{\mathbf{H}_S}^{W_{t-1}}\left[D_{KL}(Q_{\mathcal{G}_t|\mathbf{H}_s, W_{t-1}}\|P_{\tilde{N}_t|W_{t-1}})\right]\right]$, where the infimum is achieved when $P_{\tilde{N}_t|W_{t-1}} = Q_{\mathcal{G}_t|W_{t-1}}$. The KL divergence may be viewed as an estimate of the sensitivity of the full batch implicit gradient to a specific demonstration sample $\mathbf{H}_s = H_s$.*

From **Lemma 5**, every choice of $P_{\tilde{N}_t|W_{t-1}}$ gives rise to an upper bound of the MI of interest via $I(\mathcal{G}_t; \mathbf{H}_s|W_{t-1}) \leq \mathbb{E}_{W_{t-1}}\left[\mathbb{E}_{\mathbf{H}_s}^{W_{t-1}}\left[D_{KL}(Q_{\mathcal{G}_t|\mathbf{H}_s, W_{t-1}}\|P_{\tilde{N}_t|W_{t-1}})\right]\right]$. Same to Wang and Mao [51], choose an isotropic Gaussian prior $P_{\tilde{N}_t|W_{t-1}} = \mathcal{N}(\tilde{g}_t, \sigma_t^2 I_d)$, where both $\tilde{g}_t$ and $\sigma_t$ are only allowed to depend on $W_{t-1}$ under **Theorem 1**, and optimize the KL divergence in **Lemma 5** over $\sigma_t$ for a fixed $\tilde{g}_t$. Additionally, Under **Theorem 1** where we define the implicit GD trajectories of ICL, assume $C_t$ is a positive-definite matrix, for any $t \in [L]$, we have,

$$I\left(-G_t + C_t^{1/2}N_t; \mathbf{H}_s|W_{t-1} = w_{t-1}\right)$$

$$\leq \inf_{\tilde{g}_t, \sigma_t} \mathbb{E}_{\mathbf{H}_s}\left[D_{KL}\left(P_{-G_t + C_t^{1/2}N_t|W_{t-1}=w_{t-1}, \mathbf{H}_s=H_s}\|P_{-\tilde{g}_t + \sigma_t N_t|W_{t-1}=w_{t-1}}\right)\right]$$

$$= \inf_{\tilde{g}_t, \sigma_t} \mathbb{E}_{\mathbf{H}_s}\left[\frac{1}{2}\log\left(\frac{\det(\sigma_t^2 I_d)}{\det(C_t)}\right) - \frac{1}{2}d + \frac{1}{2\sigma_t^2}(G_t - \tilde{g}_t)^T I_d^{-1}(G_t - \tilde{g}_t) + \frac{1}{2\sigma_t^2}\text{tr}\left\{I_d^{-1}C_t\right\}\right] \tag{16}$$

$$= \frac{1}{2}\inf_{\tilde{g}_t, \sigma_t} \mathbb{E}_{\mathbf{H}_s}\left[\frac{1}{\sigma_t^2}(\|G_t - \tilde{g}_t\|_2^2 + \text{tr}\left\{C_t\right\}) + d\log\sigma_t^2 - d - \text{tr}\left\{\log C_t\right\}\right] \tag{17}$$

where Eq. (16) is by the KL divergence between two Gaussian distributions: $D_{KL}(p\|q) = \frac{1}{2}\left[\log\frac{\det(\Sigma_q)}{\det(\Sigma_p)} - k + (\mu_p - \mu_q)^T\Sigma_q^{-1}(\mu_p - \mu_q) + \text{tr}(\Sigma_q^{-1}\Sigma_p)\right]$, and Eq. (17) is due to the fact that $G_t^T G_t = \text{tr}(G_t G_t^T)$ and $\log\det(C_t) = \text{tr}(\log C_t)$ when $C_t$ is a positive-definite matrix.

Let $A_1(t) = \mathbb{E}_{\mathbf{H}_s}[\|G_t - \tilde{g}_t\|_2^2 + \text{tr}\left\{C_t\right\}]$ and $A_2(t) = \mathbb{E}_{\mathbf{H}_s}[\text{tr}\left\{\log C_t\right\}]$ and fix $\tilde{g}_t$ can rewrite Eq. (17) as,

$$\frac{1}{2}\inf_{\sigma_t \geq 0} \frac{1}{\sigma_t^2}A_1(t) + d\log\sigma_t^2 - d - A_2(t)$$

$$= \frac{1}{2}d\log\frac{A_1(t)}{d} - \frac{1}{2}A_2(t),$$

where the optimal $\sigma^* = \sqrt{\frac{A_1(t)}{d}}$ when we take the derivative of $\sigma_t$. Then combine **Lemma 2** and **Lemma 3**, we can finish the proof of the following lemma.

**Lemma 6.** *Under the conditions of **Theorem 1**, assume the implicit gradient noise covariance $C_t$ is a positive-define matrix, the loss $\ell(w, \mathbf{h})$ is R-subGaussian for any $w \in \mathcal{W} \in \mathbb{R}^d$. For any $t \in [L]$, let $\tilde{g}_t$ be any constant vector for a given $w_{t-1}$, then*

$$\widetilde{error} \leq \sqrt{\frac{R^2}{N} \sum_{t=1}^{L} \mathbb{E}_{W_{t-1}} \left[ d \log(A_1(t)/d) - A_2(t) \right]}$$

$$= \sqrt{\frac{R^2}{N} \sum_{t=1}^{L} \mathbb{E}_{\mathbf{W}_{t-1}}^{\mathbf{H}_s} \left[ d \log \left( \frac{\left\| \mathbf{vec}(\Delta\mathbf{W}_t(1 + \sum_{j=1}^{t-1} \mathbf{G}_j)) - \tilde{g}_t \right\|_2^2 + \mathrm{tr}\{C_t\}}{d} \right) - \mathrm{tr}\{\log C_t\} \right]},$$

*where $A_1(t) = \mathbb{E}_{W_{t-1}}^{\mathbf{H}_s} \left[ \|G_t - \tilde{g}_t\|_2^2 + \mathrm{tr}\{C_t\} \right], A_2(t) = \mathbb{E}_{W_{t-1}}^{\mathbf{H}_s} \left[ \mathrm{tr}\{\log C_t\} \right], \mathbf{vec}(\mathbf{G}_t) = G_t, \mathrm{tr}\{\cdot\}$ denotes the trace of a matrix and $\mathbb{E}_Y^X$ is the conditional expectation.*

Further, if we let $\tilde{g}_t = 0$, reverse the process of flattening the weight matrix into a vector form, and by Eq. (19) we obtain

$$\left\| \mathbf{vec}(\Delta\mathbf{W}_t(1 + \sum_{j=1}^{t-1} \mathbf{G}_j))) - \tilde{g}_t \right\|_2^2 + \mathrm{tr}\{C_t\} = \left\| \Delta\mathbf{W}_t(1 + \sum_{j=1}^{t-1} \mathbf{G}_j) \right\|_F^2 + \mathrm{tr}\{C_t\}$$

$$\leq \|\Delta\mathbf{W}_t\|_F^2 \cdot \left\| 1 + \sum_{j=1}^{t-1} \mathbf{G}_j \right\|_F^2 + \mathrm{tr}\{C_t\}.$$

Then plug everything into **Lemma 6**, we conclude the proof.

### A.6 Examples of Remark 4

First, we will present several expressions that will be utilized later. For any matrix $\mathbf{A}$ and $\mathbf{B}$, we have,

$$\|\mathbf{vec}(\mathbf{A}) \otimes \mathbf{vec}(\mathbf{B})\|_F = \sqrt{\sum_i \sum_j |\mathbf{vec}(\mathbf{A})_i \mathbf{vec}(\mathbf{B})_j|^2} = \max_i |\sigma_i(\mathbf{vec}(\mathbf{A}) \otimes \mathbf{vec}(\mathbf{B}))| \quad (18)$$

$$\|\mathbf{AB}\|_F \leq \|\mathbf{A}\|_F \|\mathbf{B}\|_F \quad (19)$$
$$\|\mathbf{A} + \mathbf{B}\|_F \leq \|\mathbf{A}\|_F + \|\mathbf{B}\|_F \quad (20)$$

Same in **Lemma 1**, $\Delta\mathbf{W}_{icl} = \left( \sum_{i=1}^{N} \mathbf{W}_V \mathbf{h}_i \otimes \mathbf{W}_K \mathbf{h}_i \right) \mathbf{W}_Q$, let $r$ represents the remained rank and $\delta$ represents the potential noise consisting of parts with small singular values. Some SVD controlling the norm of $\Delta\mathbf{W}_{icl}$ examples are as follows.

**Example 2** (Prune $\mathbf{W}_Q$). *Suppose we decompose $\mathbf{W}_Q$ by SVD, $\mathbf{W}_Q = \mathbf{W}_{Q_r} + \delta_Q = \mathbf{U}_{:r}^Q \mathbf{\Sigma}_{:r}^Q (\mathbf{V}_{:r}^Q)^T + \delta_Q$, let $\mathbf{z} \triangleq \left( \sum_{i=1}^{N} \mathbf{W}_V \mathbf{h}_i \otimes \mathbf{W}_K \mathbf{h}_i \right)$ for simplicity.*

$$\|\Delta\mathbf{W}_{icl}\|_F = \left\| \mathbf{z}\mathbf{U}_{:r}^Q \mathbf{\Sigma}_{:r}^Q (\mathbf{V}_{:r}^Q)^T + \mathbf{z}\delta_Q \right\|_F$$

$$\leq \|\mathbf{z}\mathbf{W}_{Q_r}\|_F + \|\mathbf{z}\delta_Q\|_F,$$

*where the inequality part takes advantage of the Eq. (20), so the upper bound on $\|\Delta\mathbf{W}_{icl}\|_F$ is decreasing when using SVD.*

**Example 3** (Prune $\mathbf{W}_V$ or $\mathbf{W}_K$). *Suppose we decompose $\mathbf{W}_V$ by SVD, $\mathbf{W}_V = \mathbf{W}_{V_r} + \delta_V = \mathbf{U}_{:r}^V \mathbf{\Sigma}_{:r}^V (\mathbf{V}_{:r}^V)^T + \delta_V$, and consider the upper bound defined in **Example 1**,*

$$UB(\|\Delta\mathbf{W}\|_F^2) = \sum_{i=1}^{N} \|\mathbf{W}_V \mathbf{h}_i \otimes \mathbf{W}_K \mathbf{h}_i\|_F^2 \|\mathbf{W}_Q\|_F^2$$

$$\geq \sum_{i=1}^{N} \|\mathbf{W}_{V_r} \mathbf{h}_i \otimes \mathbf{W}_K \mathbf{h}_i\|_F^2 \|\mathbf{W}_Q\|_F^2, \quad (21)$$

*where Eq. (21) is by Eq. (18), so the upper bound on $\|\Delta \mathbf{W}_{icl}\|_F$ is also decreasing. The same is true for decomposing $\mathbf{W}_K$.*

**Example 4** (Prune $\mathbf{W}_{MLP}$). *Suppose we decompose $\mathbf{W}_{MLP}$ by SVD, $\mathbf{W}_{MLP} = \mathbf{W}_{MLP_r} + \delta_{MLP} = \mathbf{U}_{:r}\mathbf{\Sigma}_{:r}\mathbf{V}_{:r}^T + \delta_{MLP}$, let $\mathbf{z} \triangleq \left(\sum_{i=1}^N \mathbf{W}_V \mathbf{h}_i \otimes \mathbf{W}_K \mathbf{h}_i\right)\mathbf{W}_Q$ for simplicity.*

$$\left\|\Delta \mathbf{W}_{icl}''\right\|_F = \left\|\mathbf{U}_{:r}\mathbf{\Sigma}_{:r}\mathbf{V}_{:r}^T\mathbf{z} + \delta_{MLP}\mathbf{z}\right\|_F$$
$$\leq \left\|\mathbf{W}_{MLP_r}\mathbf{z}\right\|_F + \left\|\delta_{MLP}\mathbf{z}\right\|_F,$$

*where recall Appendix A.3, $\Delta \mathbf{W}_{icl}'' = \mathbf{W}_{MLP}\left(\sum_{i=1}^N \mathbf{W}_V \mathbf{h}_i \otimes \mathbf{W}_K \mathbf{h}_i\right)\mathbf{W}_Q$. And the inequality part takes advantage of the Eq. (20), so the upper bound on $\left\|\Delta \mathbf{W}_{icl}''\right\|_F$ is decreasing when using SVD.*

# B More Discussions

## B.1 Explanation of the Mask Matrix

Our notation here follows the related work [2], which explains: Note that the prompt is asymmetric since the label for $\mathbf{x}_{N+1}$ is excluded from the input. To reflect this asymmetric structure, the mask matrix $\mathbf{M}$ is included in the attention. More specifically, if you pay attention to the $(N+1)$-th item, $\mathbf{M}$ is supposed to represent a causal mask (For $\mathbf{H} \in \mathbb{R}^{(dout+din)\times(N+1)}$, $\mathbf{HM} = (\mathbf{h}_1, ..., \mathbf{h}_N, \mathbf{h}_{N+1})\mathbf{M} = (\mathbf{h}_1, ..., \mathbf{h}_N, 0) = \mathbf{H}_s$). Besides, this mask method is used in GLM training [13]: Part A tokens can attend to each other, but cannot attend to any tokens in B. Part B tokens can attend to Part A and antecedents in B, but cannot attend to any subsequent tokens in B. So it is reasonable for Eq.(1) and Eq.(2) use the same mask.

## B.2 Definitions of Deep and Shallow

Indeed, there is no universally accepted definition for the terms "deep" and "shallow" as their interpretation can be subjective and dependent on the reference frame (e.g., model size). Intuitively, in this paper, our definition is similar to that of work [50]: "shallow" layers refer to those closer to the input, while "deep" layers are closer to the output. In Figure 1 and Figure 2 (Section 2), "shallow" typically denotes the first few layers, and "deep" denotes the last few layers of the network.

## B.3 The Effect of Pruning Only a Single Module

In our theory, the example (pruning only $\mathbf{W}_K$ or $\mathbf{W}_V$) is provided in Appendix A.6, demonstrating how weight pruning can affect the norm of $[\Delta \mathbf{W}_t]_1^L/[\mathbf{G}_t]_t^L$. Thus, it may confer advantages on the performance of Transformers in ICL inference. However, in **Theorem 2** (**Remark 5**), it also suggests that modifications to the shallow layers have a less steady impact. Additionally, please review the supplementary experimental results provided below. (We choose key projection matrix $\mathbf{W}_K$ and select the 3-layer (large matrix condition number in Figure 3) of GPT-J-6B, others are the same to Section 2.1).

| Task Name | Optimal $\xi^*$ | Test Accuracy/F1 Improve |
|-----------|-----------------|--------------------------|
| SST-2 | 0.0 | 0.7828 - 0.7828 |
| RTE | 0.5 | 0.5413 - 0.5413 |
| COPA | 0.995 | 0.53 - 0.54 |

## B.4 Why Optimal Clipping Rate Varies?

As we mentioned in Section 3.3 and Section C.4, the implicit gradients produced by Transformers in practical applications are noisy due to factors such as the extent of model pre-training and data characteristics (e.g., ICL shot number/task difficulty). Therefore, $[\Delta \mathbf{W}_t]_1^L / [\mathbf{G}_t]_t^L$ in **Theorem 1** have varies noise. That is why optimal $\xi$ varies.

**Gradient quality derived from context (i.e., $\mathbf{G}_t$).**

- In **Theorem 1** (Remark 2), $\mathbf{G}_t$ is only dependent on $\mathbf{W}_{t-1}$ and $\mathbf{H}_s$, this is consistent with gradient descent in terms of relevance (Conventionally, gradients in training are only related to the current parameters and the training samples).

- In real-world training scenarios, SGD computes gradient by selecting a small batch of samples per iteration. This approach approximates the true gradient, inherently introducing noise. Similarly, in In-Context Learning, a small subset of samples (context examples) is used to generate implicit gradient (i.e., $\mathbf{G}_t$), which also results in the introduction of noise.

## B.5 Apply the Same Clipping Rate to Other Datasets

As we mentioned in Appendix B.4, $[\Delta \mathbf{W}_t]_1^L / [\mathbf{G}_t]_t^L$ in **Theorem 1** exhibit varying levels of noise, causing the optimal clipping rate to vary among different tasks, as it is dependent on the specific task and data. So if we apply the clipping rate (0.95) as used in SST-2 to other datasets, the model performance can either improve or deteriorate. Additionally, It is possible to conduct a certain number of experiments to find a range of optimal clipping rate that is broadly applicable.

## B.6 What Would Happen if the Layer Was Dropped Entirely?

Dropping the layer could be the best option specifically for optimizing generalization error. Here's a detailed analysis:
(1) In our theoretical framework, we model each layer of the Transformer do a single iteration of implicit gradient descent (ICL) in **Theorem 1**. This scientific analysis references [46, 3, 10].
(2) In **Theorem 2**, L-layer Transformer:

Expected generalization error = population risk ($L_\mu$) - empirical risk ($L_{\mathbf{H}_s}$)

Expected generalization error = $\sqrt{\frac{R^2}{N} \sum_{t=1}^{L} d \log(\|\Delta \mathbf{W}_t\|_F^2 \|\mathbf{G}_t\|_F^2)}$ (only show the main part). If you drop the entire layer, it will change from $\sum_{t=1}^{L}$ to $\sum_{t=1}^{L-1}$. Therefore, dropping the layer may in fact be the best option for generalization error.

(3) However, according to the traditional statistical learning viewpoint, performance can be bounded by the sum of optimization error and generalization error (Please see **Remark 6** for "How should **Theorem 2** be interpreted?").

Thus, during the pruning process, there is a trade-off between optimization and generalization. Therefore, as pruning increases, the model's performance tends to first improve and then decline (the drop-layer method [19] does not harm but also does not improve), as demonstrated in the experiments shown in Figure 1.

In conclusion, dropping the entire layer can be a potential method (best option for generalization error) in our theoretical framework, but it may not necessarily be the best option for model performance.

## C  Extension to Experiments

**Computational resources.** We use a single NVIDIA GeForce RTX 3090 GPU and most tests run take 1-5 hours depending on dataset size and context length.

## C.1 Prompts

Table 1: The prompts of the datasets used in our experiments. Here regard the types of tasks: classification, multiple-choice as (cls.), (mch.). <> represents input from the dataset. For (mch.) tasks, we put in different candidates in the prompt and calculate the average log-likelihood of each candidate, and select the candidate with the highest score.

| Dataset | Type | Prompt |
|---------|------|--------|
| SST-2 | (cls.) | text: <text> sentiment: <label> |
| AGNEWS | (cls.) | text: <text> classification: <label> |
| EMOC | (cls.) | text: <text> sentiment: <label> |
| MRPC | (cls.) | sentence1: <sentence1> sentence2: <sentence2> label : <label> |
| RTE | (cls.) | <premise> Does this mean that <hypothesis> is true? select Yes or No? <label> |
| CB | (cls.) | Suppose <premise> Can we infer that <hypothesis>? Yes, No, or Maybe? <label> |
| COPA | (mch.) | <premise><question><choice> |

## C.2 Algorithm1

---
**Algorithm 1** Search the Optimal Clipping Rate for Downstream Tasks

---
**Require:** Pretrained model $\mathcal{M}$, dataset $\mathcal{D}$, predefined clipping rate candidate set $\mathcal{C}$,
predefined module $\mathcal{O} \in [\text{ATTN, MLP}]$
Split $\mathcal{D}$ into ICL demonstration sample set $\mathcal{H}_s$, validation set $\mathcal{V}$, and test set $\mathcal{T}$
Compute condition numbers for all layers in $\mathcal{M}$
Select layers $\mathcal{L}$ with top-$k$ largest condition numbers given $\mathcal{O}$
Select the largest layer number $l$ from $\mathcal{L}$
Initialize $\xi^* = 0$
Initialize $score^* = 0$
**for** each $\xi$ in $\mathcal{C}$ **do**
    $\mathcal{M}^{'} = Clip(\mathcal{M}, l, \xi, \mathcal{O})$
    $score = Evaluate(\mathcal{M}^{'}, \mathcal{V}, \mathcal{H}_s)$
    **if** $score > score^*$ **then**
        $\xi^* = \xi$
        $score^* = score$
    **end if**
**end for**
$\mathcal{M}^* = Clip(\mathcal{M}, l, \xi^*, \mathcal{O})$
$test\ score = Evaluate(\mathcal{M}^*, \mathcal{T}, \mathcal{H}_s)$
Output $test\ score$ on $\mathcal{T}$

---

**The details.** Firstly, the details of the algorithm can be reviewed in the code provided. Specifically, the set of clipping rate candidates is predefined. In our study, the clipping rate candidates are set as shown in Figures 1 and 2: [0, 0.1, 0.5, 0.75, 0.9, 0.95, 0.99, 0.995]. Besides, we analyze the impact of different hyperparameters through comparative experiments (details in Section 2 and Section 3.4), as detailed below. (1) Clipping rate $\xi$:We search for the optimal $\xi$ in the predefined clipping rate candidates. (2) Predefined module $\mathcal{O}$: The module containing the target pruning weights, which can be chosen from $[k\_proj, q\_proj, v\_proj, out\_proj, fc\_in, fc\_out, all, mlp, attn]$. (3) Selected layer $l$ : The layer containing the target pruning weights. For examlpe: In Section 2, we mainly focus on comparing the impact of weight pruning to the **first two and the last two layers** of the model. (4) ICL shot number : The demonstration number in ICL, we analyze the effect of different ICL shot numbers in Section 2.2.

## C.3 Dataset Details

For experiments, we consider numerous classification datasets, including: SST-2 [39], AGNEWS [58], EMOC [8], MRPC [12], RTE [5], CB [11]. We also include a multiple-choice dataset COPA [18].

**SST-2.** SST-2 focuses on sentiment analysis, specifically the task of determining the sentiment of movie reviews. The dataset consists of sentences labeled as having a positive or negative sentiment. These sentences are divided into a training set, a validation set, and a test set. The training set includes about 67,349 sentences, whereas the validation set contains around 872 sentences. The test set comprises approximately 1,821 sentences without labels. We random sample $N(10)$ data from the train set as demonstration sample, 8,000 data from the train set as validation set to search the optimal clipping rate, the rest 59,000+ data as test set.

**AGNews.** The AGNews dataset is a collection designed for text classification tasks, specifically for news categorization. It consists of news articles gathered from the AG's corpus of news on the web, which is categorized into four main topics: World, Sports, Business, and Science/Technology. In terms of structure, the AGNews dataset comprises approximately 120,000 training samples and 7,600 test samples. We random sample $N(8)$ data from the train set as demonstration sample, 8000 data from the train set as validation set to search the optimal clipping rate, the 7,600 test samples as test set.

**EmoC.** The EmoC dataset focuses on contextual emotion detection in text. It consists of short text conversations extracted from three-turn English Tweets. The dataset is annotated with four emotion labels: happiness, sadness, anger, and others, we relabel the dataset with happiness and others. The training set includes 30,160 samples, while the test set comprises 5,509 samples. We random sample $N(10)$ data from the train set as demonstration sample, 5,000 data from the train set as validation set to search the optimal clipping rate, the 5,509 test samples as test set.

**MRPC.** The MRPC dataset focuses on paraphrase detection in text. It consists of sentence pairs automatically extracted from online news sources. The dataset is annotated with binary labels indicating whether the sentences are paraphrases of each other. The training set includes 3,668 pairs, the validation set includes 408 pairs, while the test set comprises about 1,725 pairs. We random sample $N(10)$ data from the train set as demonstration sample, the 408 pairs from validation set to search the optimal clipping rate, the 1,725 test samples as test set.

**RTE.** Recognizing Textual Entailment (RTE) task, which involves assessing the relationship between a pair of sentences—a premise and a hypothesis. The objective of the task is to determine whether the hypothesis can be logically inferred from the premise. Specifically, the model must evaluate whether the relationship between the two sentences is one of "entailment," where the content of the hypothesis is directly or indirectly supported by the premise, or "non-entailment," which includes both contradiction and neutrality. In RTE from SuperGLUE [47], the training set includes about 2,490 items, whereas the validation set contains around 277 items. We random sample $N(10)$ data from the train set as demonstration sample, the remain items from train set to search the optimal clipping rate, the 277 validation samples as test set.

**CB.** The CommitmentBank (CB) dataset is part of the SuperGLUE benchmark and focuses on textual entailment with an emphasis on pragmatic inference. The task involves determining whether a hypothesis can be logically inferred from a given text, which in this dataset, typically comprises a premise followed by a hypothesis. The training set includes about 250 items, whereas the validation set contains around 56 items. We random sample $N(15)$ data from the train set as demonstration sample, the remain 235 items from train set to search the optimal clipping rate, the 56 validation samples as test set.

**COPA.** The Choice of Plausible Alternatives (COPA) dataset is specifically designed to evaluate causal reasoning abilities in natural language processing models. This dataset presents a task where models must determine causal relationships within simple scenarios. Each question in COPA consists of a premise and two possible choices, one of which is the correct cause or effect of the premise. COPA's dataset is relatively straightforward and consists of 500 questions split evenly into training (400) and validation (100) sets. We random sample $N(10)$ data from the train set as demonstration sample, the 200 items from train set to search the optimal clipping rate, the 100 validation samples as test set.

## C.4 Noise Discussion of the Implicit Gradient

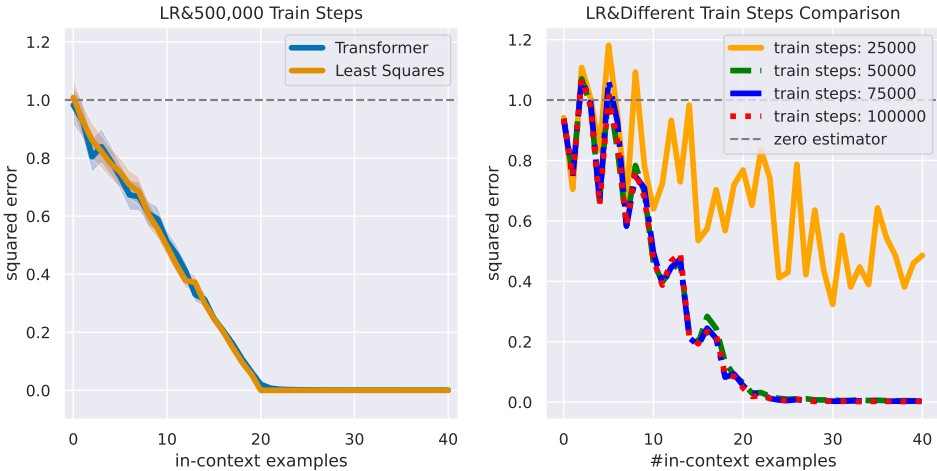

(a) Trained Transformer performs comparably to the optimal least squares estimator

(b) The effect of different train steps on Trained Transformer

Figure 5: Evaluating the trained Transformer on in-context learning linear functions. (a) Garg et al. [16] consider the class of linear functions $\mathcal{F} = \{f|f(x) = w^T x, w \in \mathbb{R}^d\}$, in $d$ dimensions where $d = 20$. They sample $x_1, \ldots, x_k, x_{\text{query}}$, and $w$ independently from the isotropic Gaussian distribution $N(0, I_d)$. They then compute each $y_i = w^T x_i$ and construct the prompt as $P = (x_1, y_1, x_2, y_2, \ldots, x_k, y_k, x_{\text{query}})$. This figure plots the normalized squared error of the Transformer $((M(P) - w^T x_{\text{query}})^2 / d)$, the errors are normalized so that the trivial zero estimator achieves an error of 1 (dashed line). Besides, when the number of in-context examples reaches the problem dimension $d$ (here 20), least squares achieves 0 error while the Transformer achieves an error of 0.02. (b) We follow the same setting of [16] in (a) to compare the Trained Transformer with different train steps.

As mentioned in Garg et al. [16]'s work, the trained model is able to learn unseen linear functions from in-context examples with performance comparable to the optimal least squares estimator, this can be seen in Figure 5a. More importantly, the number of in-context examples (shot number) plays a significant role. In this case, the error approaches zero only when the shot number is greater than or equal to $d$. This indicates that the implicit gradients of ICL are influenced by the shot number, and there exists a threshold $N$, when the actual shot number $b$ is below this threshold, the implicit gradients are noisy. On the other hand, following the same setting of [16] in figure 5a, we compare the performance of the Trained Transformer at different train steps as shown in figure 5b. This indicates that the performance of the model varies with the number of train steps, which also means that the implicit gradients of ICL generated by the model are influenced by the extent of its pre-training.

In conclusion, actual implicit gradient descent involves noise, which primarily stems from the shot number and the degree of model pre-training.

## C.5 The Detailed Results of Algorithm 1

Table 2: The results of Algorithm 1 on SST-2

| Model Name | Layer Number | Module Name | Optimal $\xi^*$ | Test Acc Improve |
|---|---|---|---|---|
| GPT-J-6B | 26 | MLP | 0.95 | 0.7527 - **0.8437**(↑) |
| GPT-J-6B | 27 | ATTN | 0.995 | 0.7527 - 0.7642(↑) |
| LLAMA2-7B | 30 | MLP | 0.99 | 0.9228 - 0.9257(↑) |
| LLAMA2-7B | 30 | ATTN | 0.95 | 0.9228 - **0.9287**(↑) |

Table 3: The results of Algorithm 1 on AGNEWS

| Model Name | Layer Number | Module Name | Optimal $\xi^*$ | Test Acc Improve |
|---|---|---|---|---|
| GPT-J-6B | 26 | MLP | 0.1 | 0.76434 - 0.76947($\uparrow$) |
| GPT-J-6B | 27 | ATTN | 0.95 | 0.76434 - **0.77026**($\uparrow$) |
| LLAMA2-7B | 30 | MLP | 0.995 | 0.77026 - **0.84881**($\uparrow$) |
| LLAMA2-7B | 30 | ATTN | 0.1 | 0.77026 - 0.77039($\uparrow$) |

Table 4: The results of Algorithm 1 on EmoC

| Model Name | Layer Number | Module Name | Optimal $\xi^*$ | Test Acc Improve |
|---|---|---|---|---|
| GPT-J-6B | 26 | MLP | 0.1 | 0.69032 - **0.74278**($\uparrow$) |
| GPT-J-6B | 27 | ATTN | 0.5 | 0.69032 - 0.69758($\uparrow$) |
| LLAMA2-7B | 30 | MLP | 0.1 | 0.77110 - **0.79106**($\uparrow$) |
| LLAMA2-7B | 30 | ATTN | 0.1 | 0.77110 - 0.76910($\downarrow$) |

Table 5: The results of Algorithm 1 on MRPC

| Model Name | Layer Number | Module Name | Optimal $\xi^*$ | Test Acc Improve |
|---|---|---|---|---|
| GPT-J-6B | 26 | MLP | 0.995 | 0.66492 - 0.66492($-$) |
| GPT-J-6B | 27 | ATTN | 0.995 | 0.66492 - 0.66492($-$) |
| LLAMA2-7B | 30 | MLP | 0.5 | 0.66666 - **0.67536**($\uparrow$) |
| LLAMA2-7B | 30 | ATTN | 0.995 | 0.66666 - 0.66608($\downarrow$) |

Table 6: The results of Algorithm 1 on RTE

| Model Name | Layer Number | Module Name | Optimal $\xi^*$ | Test Acc Improve |
|---|---|---|---|---|
| GPT-J-6B | 26 | MLP | 0.99 | 0.56884 - **0.57971**($\uparrow$) |
| GPT-J-6B | 27 | ATTN | 0 | 0.56884 - 0.56884($-$) |
| LLAMA2-7B | 30 | MLP | 0.75 | 0.56159 - 0.57608($\uparrow$) |
| LLAMA2-7B | 30 | ATTN | 0.9 | 0.56159 - **0.57971**($\uparrow$) |

Table 7: The results of Algorithm 1 on CB

| Model Name | Layer Number | Module Name | Optimal $\xi^*$ | Test Acc Improve |
|---|---|---|---|---|
| GPT-J-6B | 26 | MLP | 0.95 | 0.58181 - **0.67272**($\uparrow$) |
| GPT-J-6B | 27 | ATTN | 0 | 0.58181 - 0.58181($-$) |
| LLAMA2-7B | 30 | MLP | 0.95 | 0.83636 - 0.85454($\uparrow$) |
| LLAMA2-7B | 30 | ATTN | 0.99 | 0.83636 - **0.87272**($\uparrow$) |

Table 8: The results of Algorithm 1 on COPA

| Model Name | Layer Number | Module Name | Optimal $\xi^*$ | Test F1 Improve |
|---|---|---|---|---|
| GPT-J-6B | 26 | MLP | 0.95 | 0.58 - **0.59**($\uparrow$) |
| GPT-J-6B | 27 | ATTN | 0.99 | 0.58 - 0.58($-$) |
| LLAMA2-7B | 30 | MLP | 0.5 | 0.57 - **0.62**($\uparrow$) |
| LLAMA2-7B | 30 | ATTN | 0.99 | 0.57 - 0.56($\downarrow$) |

