# OpenReview forum: "Enhancing In-Context Learning Performance with just SVD-Based Weight Pruning: A Theoretical Perspective"
_NeurIPS.cc/2024/Conference — NeurIPS 2024 poster_

### Official Review · Reviewer_CqwV · 2024-07-05

**Soundness:** 3
**Presentation:** 2
**Contribution:** 3
**Rating:** 5
**Confidence:** 3

**Summary:**

The authors provide a theoretical perspective on the stability of in context learning via implicit gradient descent trajectories. Ultimately, the analysis suggests that high condition numbers of the weight matrices belonging to layers with a high index can be pruned in order to achieve a model which performs better on ICL tasks.

**Strengths:**

- In context learning is important, and something which has not been studied as deeply as other topics of ML due to the recent rise of transformers and ICL in general.
- The method intuitively makes sense and is something which can be conditionally tuned after training based on specific tasks if a validation set is available.

**Weaknesses:**

- It would be good to define deep and shallow, as these are subjective terms depending on the reference frame.
- Figure 1 cpation says: "We operate on the whole of MLP or ATTN." What does this mean?
- If as figure 1 states, you can clip 99.5% of the original weights, what happens if you just drop that layer entirely? Recent work has shown that the deeper layers can be completely dropped without much effect. [1]
  - I cannot see much benefit gained from pruning part of the weights with SVD when it seems that the in nearly all cases, the benefit can be had by dropping the layer entirely.

- Is the mask on L138 supposed to represent a causal mask? If so, I do not think the notation is correct, as the Identity matrix would only have $N$ binary values which is much less than is needed for a causal mask.
- How can equation 1 and 2 use the same mask?
- Example 1 appears to be incorrect:
  - There is no parentheses around $W_{V_r}^k + \delta_V h_i^{k-1}$ in the first line.
  - The triangle inequality seems to say that line 2 $\geq$ line 1
  - Given the above, I do not see what conclusion can be drawn from this equation.
  - Have I missed something here?

**Questions:**

- I don't believe the clipping process was adequately explained. Once the SVD operation is done, do you clip starting from the largest singular value? or starting from the smallest?

**Limitations:**

The limitations have been discussed.

---

> ### Author Rebuttal · Authors · 2024-08-02
>
> We appreciate your response on finding our work thorough and informative. Below we address specific questions.
>
> > **W1:** It would be good to define deep and shallow, as these are subjective terms depending on the reference frame.
>
> > **A:** Indeed, there is no universally accepted definition for the terms "deep" and "shallow"  as their interpretation can be subjective and dependent on the reference frame (eg. model size). Intuitively,  in this paper, our definition is similar to that of Work [1]: “shallow”  layers refer to those closer to the input, while “deep”  layers are closer to the output. In Figure 1 and Figure 2 (**Section 2**), "shallow" typically denotes the first few layers, and "deep" denotes the last few layers of the network.
> >
> >
> > [1] Lean Wang et al. Label Words are Anchors: An Information Flow Perspective for Understanding In-Context Learning. EMNLP 2023.
>
> > **W2:** Figure 1 cpation says: "We operate on the whole of MLP or ATTN." What does this mean?
>
> > **A:** It refers to pruning all the weight matrices in these modules. The striking model architecture Transformer block consists of an attention (ATTN) layer and a feed-forward (MLP) layer. As mentioned in **Section 2**, the attention (ATTN) layers consist of key, query, value, and output matrices in both GPT-J-6B and LLAMA2-7B. The MLP layers in GPT-J-6B include input and output matrices, while in LLAMA2-7B, they are composed of up, gate, and down matrices.
> >
>
> > **W3:** If as figure 1 states, you can clip 99.5% of the original weights, what happens if you just drop that layer entirely? Recent work has shown that the deeper layers can be completely dropped without much effect. [1]
> >
> > - I cannot see much benefit gained from pruning part of the weights with SVD when it seems that the in nearly all cases, the benefit can be had by dropping the layer entirely.
>
> > **A:** Thank you for your good question. To provide a thorough and accurate answer, could you please share the reference for [1] that you mentioned? Understanding the specifics of this recent work would help us address your inquiry more effectively.
> >
> > Regarding your question, I have two key points to consider:
> >
> > (i) Even though we can "clip 99.5% of the original weights," the remaining weights are still significant and play a crucial role. This is because the remaining 0.5% corresponds to the larger singular values after performing SVD, which are vital for maintaining the model's performance.
> >
> > (ii) As stated in **Theorem 1**, from the perspective of ICL (In-Context Learning) and gradient descent, each attention layer corresponds to an implicit gradient descent step (**L** layers with Implicit Gradient Descent Trajectories $[\Delta W_t]_1^L$/$[G_t]_t^L$).
> >
> > Therefore, if we just drop the entire layer, it will result in two main issues: (a) It will become challenging to adjust the weights for another downstream task. (b) The implicit gradient update will be reduced by one step, generally leading to a decline in model performance.
> >
> > Thank you again for your understanding.
>
> > **W4:** Is the mask on L138 supposed to represent a causal mask? If so, I do not think the notation is correct, as the Identity matrix would only have $N$ binary values which is much less than is needed for a causal mask.
> >
> > **W5:** How can equation 1 and 2 use the same mask?
>
> > **A:** Our notation here follows the related work [1], which explains: Note that the prompt is asymmetric since the label for $x_{N+1}$ is excluded from the input. To reflect this asymmetric structure, the mask matrix $M$ is included in the attention.
> >
> > More specifically, if you pay attention to the $(N+1)$-th item, L138 is supposed to represent a causal mask.
> >
> >  (For $H\in \mathbb{R}^{(dout+din)\times(N+1)}$, $HM=(h_1,..,h_N,h_{N+1})M=(h_1,..,h_N,0)=H_s$)
> >
> > Besides, this mask method is used in GLM training [2]: Part A tokens can attend to each other, but cannot attend to any
> tokens in B. Part B tokens can attend to Part A and antecedents in B, but cannot attend to any subsequent tokens in B. So it is reasonable for equation 1 and 2 use the same mask.
> >
> > [1] wangjun Ahn et al. Transformers learn to implement preconditioned gradient descent for in-context learning. NeurIPS 2023.
> >
> > [2] Zhengxiao Du et al. GLM: General Language Model Pretraining with Autoregressive Blank Infilling. ACL 2022.
>
> > **W6:** Example 1 appears to be incorrect:
> >
> > - There is no parentheses around $W^k_{V_r}+ \delta_Vh_i^{k−1}$ in the first line.
> > - The triangle inequality seems to say that line 2 ≥ line 1
>
> > **A:** Thank you for pointing out the typo (parentheses). We appreciate your attention to detail and will have made the necessary corrections. However, this minor typo does not affect the final conclusion. A detailed analysis is as follows:
> >
> >  In triangle inequality, it is ture that $||\mathbf{a}||_2+||\mathbf{b}||_2\geq ||\mathbf{a}+\mathbf{b}||_2$ for any vector $\mathbf{a},\mathbf{b}$ (same dimension).
> >
> >  **But in this Example, we did not use the triangle inequality**:
> >
> > $||W^k\_{V\_r}h\_i^{k−1}+\delta\_Vh\_i^{k−1}||\_2^2=||W^k\_{V\_r}h\_i^{k−1}||\_2^2+2(W^k\_{V\_r}h\_i^{k−1})^T(\delta\_Vh\_i^{k−1})+||\delta\_Vh\_i^{k−1}||\_2^2=||W^k\_{V\_r}h\_i^{k−1}||\_2^2+||\delta\_Vh\_i^{k−1}||\_2^2$.
> >
> > Note that $W^k_{V_r}=U_{:r}\Sigma_{:r}V_{:r}^T$ and $\delta_V=U_{r:}\Sigma_{r:}V_{r:}^T$, and $U_{:r},U_{r:}$ are **orthometric** (properties of SVD).
> >
> > Therefore, $(W^k_{V_r}h_i^{k−1})^T(\delta_Vh_i^{k−1})=(h_i^{k−1})^TV_{:r}\Sigma_{:r}(U_{:r}^TU_{r:})\Sigma_{r:}V_{r:}^Th_i^{k−1}=0$, line 2 = line 1.
>
> > **Q1:** I don't believe the clipping process was adequately explained. Once the SVD operation is done, do you clip starting from the largest singular value? or starting from the smallest?
>
> > **A:** As mentioned in **Section 2**, "The optimal rank-r approximation and SVD," we retain the components corresponding to the largest r singular values. Therefore, the clipping process begins with the smallest singular values.

---

> > ### Comment · Reviewer_CqwV · 2024-08-12
> > **Thank you**
> >
> > Thank you for the response. I apologize about the missing reference, it must have erroneously been excluded. The reference is:
> >
> > [1] Gromov, A., Tirumala, K., Shapourian, H., Glorioso, P., & Roberts, D. A. (2024). The unreasonable ineffectiveness of the deeper layers. arXiv preprint arXiv:2403.17887.
> >
> > I do not understand the following response above:
> >
> > > (ii) As stated in Theorem 1, from the perspective of ICL (In-Context Learning) and gradient descent, each attention layer corresponds to an implicit gradient descent step (L layers with Implicit Gradient Descent Trajectories $[\Delta W_t]_1^L$/$[G_t]_t^L$).
> >
> > >Therefore, if we just drop the entire layer, it will result in two main issues: (a) It will become challenging to adjust the weights for another downstream task. (b) The implicit gradient update will be reduced by one step, generally leading to a decline in model performance.
> >
> > How would it be challenging to adjust the weights for another downstream task? Given the fact that many models use stochastic depth, and the above reference suggests that dropping a significant portion of the layers has a minimal effect. It would seem that pruning 99.5% of the weights might be almost identical to just skipping them altogether.
> >
> > for a), As the weights are merely skipped and not totally deleted from the model, I do not understand how this would affect other tasks directly, as they could still be used in the future if necessary.
> >
> > for b) this is an assumption that has not been backed up by any data as far as I can tell.

---

> > > ### Author Response · Authors · 2024-08-12
> > > **Thank you**
> > >
> > > >Dear Reviewer CqwV,
> > > >
> > > >Thank you for your reply and for resharing the reference.
> > >
> > > > Firstly, let's briefly analyze this reference and make some comparisons:
> > > >
> > > > **Abstract:** (you provided) We empirically study a simple **layer-pruning** strategy for popular families of open-weight pretrained LLMs, finding minimal **degradation** of performance on different question-answering benchmarks until after a large fraction (up to half) of the layers are removed. To prune these models, we identify the optimal block of layers to prune by considering similarity across layers; then, **to “heal” the damage, we perform a small amount of finetuning**......
> > > >
> > > > - **Cp1:** The pruning method in the reference you provided is '**layer-based** pruning', while ours is '**weight-based** pruning'.
> > > > - **Cp2:** The pruning method in the reference you provided causes **degradation** in model performance, whereas ours leads to **improvement**.
> > > > - **Cp3:** The pruning method in the reference you provided may require **fine-tuning to to “heal” the damage**, while ours is **gradient-free**.
> > >
> > > > Next, let's address your questions:
> > > >
> > > > **Q1:** How would it be challenging to adjust the weights for another downstream task?
> > > >
> > > >  **Qa:** As the weights are merely skipped and not totally deleted from the model, I do not understand how this would affect other tasks directly, as they could still be used in the future if necessary
> > >
> > > > As shown in **Cp1** and **Cp3**, compared to removing layers and fine-tuning, only weight pruning is easier to recover from and adapt to another task. Moreover, without the need for gradient updates, this is a significant advantage of pruning that cannot be overlooked.
> > >
> > > > **Q2:** Given the fact that many models use stochastic depth, and the above reference suggests that dropping a significant portion of the layers has a minimal effect. It would seem that pruning 99.5% of the weights might be almost identical to just skipping them altogether.
> > > >
> > > > **Qb:** this is an assumption that has not been backed up by any data as far as I can tell.
> > >
> > > > As  shown in **Cp2** , although the above reference suggests that dropping a significant portion of the layers has a minimal effect, **the effect is negtive (degradation & need finetuning to heal damage)**. In contrast, the pruning method under our theoretical framework **does not require fine-tuning and results in performance improvement**.
> > > >
> > > > Regarding what you mentioned about 'It would seem that pruning 99.5% of the weights might be almost identical to just skipping them altogether'.
> > > > Typically, the dimensions of a model number in the thousands (for instance, GPTJ is 4096). **We sincerely ask, why do you think the remaining rank (i.e.,  4096*0.5%>20) after pruning is no longer important**?
> > > >
> > >
> > > >Once again, we thank you for your time and comments.

---

> > > > ### Author Response · Authors · 2024-08-12
> > > > **Main focus of our work**
> > > >
> > > > > Additionally, please allow us to underscore the **main focus of our work**, as outlined in the global rebuttal.
> > > > >
> > > > > Our primary objective is to provide a general **theoretical framework** that reveals the underlying mechanism behind the phenomenon that SVD-based weight pruning can enhance ICL performance.  Based on our theoretical insights, we can **design new algorithms** to enhance ICL performance. Consequently, we did not compare our approach with other pruning methods. Algorithm 1 is presented solely to illustrate how theoretical analysis can guide experimental procedures effectively.
> > > > >
> > > > > ( **Reviewer Zh6a** precisely pinpointed the key points: theoretical analysis can be applied to design new algorithms like algorithm 1 in this paper）

---

> > > > > ### Comment · Reviewer_CqwV · 2024-08-13
> > > > > **Clarification**
> > > > >
> > > > > I want to be clear in stating that I never asked for a direct comparison with the work I cited. I merely suggested that the results from there may imply that a high clipping rate may suggest that there is limited usefulness in that layer, and it might be dropped entirely without harming performance at all. The work from [1] shows that even with no finetuning, 20-30% of layers can be dropped with virtually no effect on the downstream task ([1] - Figure 1).
> > > > >
> > > > > > Consequently, we did not compare our approach with other pruning methods
> > > > >
> > > > > I did not ask for a comparison.
> > > > >
> > > > > > We sincerely ask, why do you think the remaining rank (i.e., 4096*0.5%>20) after pruning is no longer important.
> > > > >
> > > > > I never stated it wasn't important, I asked what would happen if that layer was dropped entirely. It simply may be the case that clipping a layer at 100% (i.e. dropping the layer) may in fact be the best option.
> > > > >
> > > > > What is the predefined clipping rate set which is mentioned in algorithm 1? I do not see it defined anywhere within the paper.

---

> > > > > > ### Author Response · Authors · 2024-08-13
> > > > > > **Thank you for your response**
> > > > > >
> > > > > > > Dear Reviewer CqwV,
> > > > > > >
> > > > > > > Thank you for your time and clarifying your points. We deeply apologize for any misunderstanding that may have occurred. We truly value your insightful suggestion, as it provides us with an important opportunity to further clarify this matter. Below, we will address specific questions to ensure comprehensive understanding.
> > > > > >
> > > > > > >**Q1:** What would happen if that layer was dropped entirely. It simply may be the case that clipping a layer at 100% (i.e. dropping the layer) may in fact be the **best option**.
> > > > > >
> > > > > > >**A:** This is indeed a very meaningful question! You're absolutely right; dropping the layer **could be the best option specifically for optimizing generalization error**. Here's a detailed analysis:
> > > > > > >
> > > > > > >(1) In our theoretical framework, we model **each layer** of the Transformer do a **single iteration of implicit gradient descent** (ICL). This scientific analysis references [1, 2, 3].
> > > > > > >
> > > > > > >(2) In **Theorem 2**,  **L**-layer Transformer:
> > > > > > >
> > > > > > >Expected generalization error = population risk ($L_{\mu}$) - empirical risk ($L_{H_s}$)
> > > > > > >
> > > > > > >Expected generalization error = $\sqrt{\frac{R^2}{N}\sum_{t=1}^Ld\log (\lVert\Delta W_t\rVert_F^2 \lVert G_t\rVert_F^2)}$ (show the main part). If you drop the entire layer, it will
> > > > > > >
> > > > > > >change from $\sum_{t=1}^L$ to $\sum_{t=1}^{L-1}$. Therefore, dropping the layer may in fact be the **best option for generalization error**.
> > > > > > >
> > > > > > >(3) However, according to the traditional statistical learning viewpoint, performance can be **bounded by the sum of optimization error and generalization error** (Please see global rebuttal for "How should Theorem 2 be interpreted?").
> > > > > > >
> > > > > > >Thus, during the pruning process, there is a **trade-off** between optimization and generalization. Therefore, as pruning increases, the model's performance tends to **first improve and then decline** (the provided reference's method does not harm but also does not improve), as demonstrated in the experiments shown in **Figure 1**.
> > > > > > >
> > > > > > >**In conclusion, your suggestion (dropping the entire layer) can be a potential method (best option for generalization error) in our theoretical framework, but it may not necessarily be the best option for model performance.**
> > > > > > >
> > > > > > >[1] Johannes von Oswald et al. Transformers learn in-context by gradient descent. ICML 2023.
> > > > > > >
> > > > > > >[2] Ekin Akyiurek et al. What learning algorithm is in-context learning? investigations with linear models. ICLR 2023.
> > > > > > >
> > > > > > >[3] Damai Dai et al. Why can gpt learn in-context? language models secretly perform gradient descent as meta-optimizers. ACL(Findings) 2023.
> > > > > >
> > > > > > >**Q2:** What is the predefined clipping rate set which is mentioned in algorithm 1?
> > > > > >
> > > > > > > **A:** Firstly, the details of the algorithm can be reviewed in the **code** provided. Specifically, the set of clipping rate candidates is predefined. In our study, the clipping rate candidates are set as shown in **Figures 1 and 2**: [0, 0.1, 0.5, 0.75, 0.9, 0.95, 0.99, 0.995]. Besides, we analyze the impact of different hyperparameters through comparative experiments (details in **Section 2** and **Section 3.4**), as detailed below (We will provide further explanations in subsequent versions).
> > > > > > >
> > > > > > > - Clipping rate $\xi$：We search for the optimal $\xi$ in the predefined clipping rate candidates.
> > > > > > >
> > > > > > > - Predefined module $\mathcal{O}$ :
> > > > > > >
> > > > > > >   The module containing the target pruning weights, which can be chosen from
> > > > > > >
> > > > > > >   ['k_proj', 'q_proj', 'v_proj', 'out_proj', 'fc_in', 'fc_out', 'all', 'mlp', 'attn']
> > > > > > >
> > > > > > > - Selected layer $l$ : The layer containing the target pruning weights. For examlpe: In **Section 2**, we mainly focuse on comparing the impact of weight pruning to the **first two and the last two layers** of the model.
> > > > > > >
> > > > > > > - ICL shot number : The demonstration number in ICL, we analyze the effect of different ICL shot numbers in **Section 2.2**.

---

### Official Review · Reviewer_Nxqr · 2024-07-12

**Soundness:** 2
**Presentation:** 1
**Contribution:** 2
**Rating:** 5
**Confidence:** 3

**Summary:**

This paper investigates the effect of singular value decomposition (SVD)-based weight pruning on the in-context learning (ICL) performance of large language models.

The Authors show that SVD-based pruning can enhance ICL performance, with deeper layers showing more stable improvements.
They provide theoretical analysis to explain these findings, presenting implicit gradient descent trajectories for ICL and deriving generalization bounds.

Based on their insights, they propose a simple algorithm for enhancing ICL inference in downstream tasks.

**Strengths:**

- The Authors provide a theoretical analysis to explain their empirical findings, including the derivation of implicit gradient descent trajectories and generalization bounds for ICL.

- Furthermore, they propose a simple, derivative-free algorithm for enhancing ICL performance in downstream tasks, demonstrating the practical value of their theoretical insights.

**Weaknesses:**

- The theoretical analysis primarily focuses on linear attention, which may not fully capture the complexities of standard Softmax attention used in most transformer models

- The proposed algorithm is derivative-free, but the search for optimal clipping rates may still be computationally expensive for very large models or datasets

- There is a substantial lack of comparison with other pruning methods: the study focuses on SVD-based pruning but doesn't compare it with other pruning techniques, which could provide context for the method's effectiveness

- Poor language, frequent typos, and grammatical errors are significant issues in this paper. This does not help readability, and would likely be a barrier to publication in its current form.

- An essential part of the paper, which is the discussion of related works is not part of the main text. Furthermore, this discussion is prone to criticism. For example, quoting the seminal paper by Frankle and Carbin as an example of low-rank properties of neural networks is clearly misleading. I think that this discussion should be an essential part of the main text, and should also be substantially revised in order to avoid conceptual confusions.

**Questions:**

I suggest the Authors to address 1) the concerns regarding a better framing of the work in the current literature. In particular, there is a large body of evidence that is growing on the resilience of LLMs to pruning (see for example "The Unreasonable Ineffectiveness of the Deeper Layers", Gromov et al, https://arxiv.org/pdf/2403.17887, and references therein), and 2) the quality of writings by proofreading it together with a native speaker.

**Limitations:**

Limitations are discussed in the final section (4).

---

> ### Author Rebuttal · Authors · 2024-08-03
>
> Thanks so much for your time and insightful comments. Please find our point-by-point response below.
>
> > **W1:**  The theoretical analysis primarily focuses on linear attention, which may not fully capture the complexities of standard Softmax attention used in most transformer models
>
> > **A:**	Firstly, to facilitate qualitative analysis, our main text primarily focuses on the theoretical aspects of linear attention.  This is our first step on this issue, and our goal is to provide insightful conclusions under simplified conditions.
> >
> > Secondly,  a considerable portion of the works also consider from the linear attention to explore the ICL [1,2].
> >
> > However, we also discuss the standard Softmax attention setting in **Appendix B.2**. Specifically, it can be considered that the mapping function $\phi$, or rather the effect of the **Softmax** function, is to project the original features into a higher-dimensional space to capture more profound features. Subsequently, learning and meta-optimization are conducted under this **new feature space**.
> >
> > $\hat{h}\_{N+1}=h\_{N+1} + \Delta W\_{icl}^{'}\phi(W\_Q h\_{N+1}),\ \Delta W\_{icl}^{'}=\frac{1}{D^{'}} \left[\sum\_{i=1}^N (W\_VH\_s)\_i \otimes \phi(W\_KH\_s)\_i \right]$
> >
> > [1] Ekin Akyiurek et al. What learning algorithm is in-context learning? investigations with linear models. ICLR 2023.
> >
> > [2] Johannes von Oswald et al. Transformers learn in-context by gradient descent. ICML 2023.
>
> > **W2:**  The proposed algorithm is derivative-free, but the search for optimal clipping rates may still be computationally expensive for very large models or datasets
>
> > **A:**  (1) Our primary objective in this work is to provide a general theoretical framework that reveals the underlying mechanism behind the phenomenon that SVD-based weight pruning can enhance ICL performance. Based on this theory, a simple algorithm is provided to demonstrate the effectiveness of theoretical analysis in guiding experimental procedures.
> >
> > In **Theorem 2**, the **key insight** is that managing the expected generalization error by controlling the norm of $\Delta W_t/G_t$, indicating that pruning methods are not unique. Therefore, some computationally efficient pruning methods could be considered: (a) Random SVD, (b) Magnitude-based pruning [1], and so on.
> >
> > [1] Wen, W et al. Learning structured sparsity in deep neural networks. NeurIPS 2016.
>
> > **W3:**  There is a substantial lack of comparison with other pruning methods: the study focuses on SVD-based pruning but doesn't compare it with other pruning techniques, which could provide context for the method's effectiveness
>
> > **A:** Thanks for your suggestion.
> > Similar to the answer to the above **W2**, we primarily focus on theoretical analysis. And the simple algorithm is just used to validate the effectiveness of the theory, so we have not provided comparative methods. There is a possibility that better pruning methods exist.
> >
> > For instance, if we use magnitude-based pruning on matrix $A$, the $||A||_F$ will be decrease.
>
> > **W4:**  Poor language, frequent typos, and grammatical errors are significant issues in this paper. This does not help readability, and would likely be a barrier to publication in its current form.
>
> > **A:** Thank you for your feedback. We sincerely apologize for the language issues, typos, and grammatical errors present in the paper. Rest assured, we are fully committed to addressing these concerns and will do everything possible to correct the issues before resubmission. Your patience and understanding are greatly appreciated, and we are dedicated to improving the paper to meet the required standards.
>
> > **W5:**  An essential part of the paper, which is the discussion of related works is not part of the main text. Furthermore, this discussion is prone to criticism. For example, quoting the seminal paper by Frankle and Carbin as an example of low-rank properties of neural networks is clearly misleading. I think that this discussion should be an essential part of the main text, and should also be substantially revised in order to avoid conceptual confusions.
>
> > **A:**  (1) Due to page limitations, we previously omitted the detailed discussion of related works in the main text, but we will include it in future versions. (2) We acknowledge your viewpoint and will make substantial revisions accordingly. Our findings show that pruning does not necessarily worsen performance (Note that performance depends not only on generalization but also on optimization). In some datasets, it may even improve results. This aligns with previous work, which we will discuss in detail in the Related Works section, such as [1]. In future research, we will study the effects of pruning different layers and the performance variations across different tasks.
> >
> > [1] Pratyusha Sharma et al. The truth is in there: Improving reasoning in language models with layer-selective rank reduction. ICLR 2024.
>
> > **Q1:**  I suggest the Authors to address 1) the concerns regarding a better framing of the work in the current literature. In particular, there is a large body of evidence that is growing on the resilience of LLMs to pruning (see for example "The Unreasonable Ineffectiveness of the Deeper Layers", Gromov et al, https://arxiv.org/pdf/2403.17887, and references therein), and 2) the quality of writings by proofreading it together with a native speaker
>
> > **A:** Thanks. We will definitely revise the literature review (Gromov et al,and references therein) & our writings.
> >
> > However,  please let us demonstrate the advantages of our theoretical analysis (pruning weight) compared to the example (remove layer) you provided.
> > If we just remove the entire layer, it will result in two main issues: (a) It will become challenging to adjust the weights for another downstream task. (b) The implicit gradient update will be reduced by one step (**Theorem 1**), perhaps leading to a decline in model performance.

---

> ### Author Response · Authors · 2024-08-11
> **Looking forward to your reply**
>
> > Dear Reviewer Nxqr,
> >
> > We sincerely appreciate your time and effort in reviewing our manuscript and providing valuable feedback.
> >
> > As the author-reviewer discussion phase nears completion, we wish to confirm whether our responses have effectively addressed your concerns. We provided detailed responses to your concerns a few days ago and hope they have adequately resolved any issues. If you require further clarification or have any additional concerns, please do not hesitate to contact us. We are more than willing to continue our communication with you.
> >
> > Best regards.

---

> > ### Comment · Reviewer_Nxqr · 2024-08-13
> > **Reviewer reply**
> >
> > Dear Authors,
> > I apologize for being late in my reply.
> >
> > I sincerely thank the Authors for their clarification. After going through the discussion with the other reviewers, and also taking into account the level of confidence of my rewiew I decided to raise my score.

---

> > > ### Author Response · Authors · 2024-08-13
> > > **Thank you and looking forward to your reconfirmation of the score**
> > >
> > > > Dear Reviewer Nxqr,
> > > >
> > > > Thank you sincerely for your time and valuable feedback. We are grateful for your willingness to raise the score following our discussion. However, we noticed that the revised score appears to be recorded as **1 point**. Could there have been an error in this update? We would appreciate your confirmation on this matter.
> > > >
> > > > Best regards.

---

> > > > ### Comment · Reviewer_Nxqr · 2024-08-13
> > > > **Corrected**
> > > >
> > > > Dear Authors, sorry, it was a mistake. The score is now the intended one.

---

> > > > > ### Author Response · Authors · 2024-08-13
> > > > > **Sincerely thank you again**
> > > > >
> > > > > > Dear Reviewer Nxqr,
> > > > > >
> > > > > > We are deeply grateful for your positive feedback and for the time and effort you have invested in your thoughtful review. Your valuable insights have significantly enriched our work. Thank you once again for your invaluable contribution.
> > > > > >
> > > > > > Warmest regards.

---

### Official Review · Reviewer_Zh6a · 2024-07-12

**Soundness:** 3
**Presentation:** 2
**Contribution:** 3
**Rating:** 5
**Confidence:** 3

**Summary:**

This paper demonstrates that (1) SVD-based weight pruning can sometimes achieve better in-context learning performance, and (2) pruning weights in deeper layers often results in more stable outcomes compared to shallow layers. The authors explain their findings through theoretical analysis and propose an intuitive matrix condition number-based weight pruning algorithm to achieve both stable and improved performance.

**Strengths:**

This work conducts an in-depth analysis to explain the "stability" of transformer weight pruning across different layers. The framework is interesting and validated through experiments. Moreover, the theoretical analysis can be applied to design new algorithms like algorithm 1 in this paper .

**Weaknesses:**

Despite adopting various simplifications (such as using a linear attention transformer without MLP and layer normalization, treating each in-context example as a single vector, implementing attention masks for query tokens, and using meta-gradients for in-context learning) in their theoretical analysis, the results are still limited. They only explain why SVD-based weight pruning can achieve "stable" performance, leaving the more intriguing question of why transformers can achieve "better" performance with pruning unclear. Additionally, even with detailed hyperparameter tuning, the effectiveness of Algorithm 1 remains uncertain. Further details are provided in the questions section.

**Questions:**

[1] How should Theorem 2 be interpreted? It seems only provide a weak upper bound for in-context learning stableness. Can this also be applied to empirical risk, such as $L_{H_S}$ and $L_\mu$?

 [2] Theorem 2 gives the upper bound for expected generalization error. If we fix N in the constructed transformer and reduce the number of in-context examples to $N'$ in the input sequence, then we can find that while the factor $R^2/N$ remains unchanged in theorem 2 , $\Delta W_t$ will change from $(\sum_{i=1}^N)…$ to $(\sum_{i=1}^{N'})…, where $ ($N' < N$). Based on the analysis across different layers, could this mean that fewer context examples are more robust for SVD weight pruning?

[3] Note that in fig-3, large matrix condition numbers can exist in some modules of shallow layers, such as the attention key (K) in GPT-J-6B . What would be the effect of pruning only a single module in a shallow layer (e.g., the key projection matrix) rather than pruning the entire attention module (including Q, K, and V)?

 [4] In C.5, it's noted that the optimal clipping rate is sometimes very small and varies across datasets. What would happen if we apply the same clipping rate (e.g., 0.95) as used in SST-2 to other datasets?

**Limitations:**

See Weaknesses and Questions. Besides, there seem many mistakes on pages 7 and 8, for example, the equation between line 228 and 229, and the equation for about F-norm in lines 230,231,232. These inaccuracies cast doubt on the overall reliability of the paper's findings. If there are any misunderstandings on my part, please point them out, and I will reconsider my evaluation of this work.

---

> ### Author Rebuttal · Authors · 2024-08-02
>
> Thank you for your review, and for your thoughtful comments and questions. They will certainly help improve the revised paper. Please see our response below.
>
> > **A to Weaknesses:** Thanks to Reviewer Zh6a for raising the issue, which gives us the opportunity to clarify this matter.
> >
> > (1) Our goal is to provide insightful conclusions under simplified conditions. However, we also discuss the standard Softmax attention setting in **Appendix B.2** and feed-forward (MLP) layers in **Appendix B.3**.
> >
> > (2) The goal of this paper is to study ICL from a generalization perspective. According to the traditional statistical learning viewpoint, performance can be defined by the sum of optimization error and generalization error. This paper reveals that SVD decomposition is beneficial for generalization error, but does not analyze its impact on optimization. Given that theoretical analysis of ICL is rapidly developing, the impact of weight pruning on optimization has not yet been fully explored. We will consider this aspect in our future work.
>
> > **A to Q1** (How should Theorem 2 be interpreted? Can this also be applied to empirical risk ?):
> >
> > Expected generalization error (**Theorem 2**) = population risk ($L\_{\mu}$) - empirical risk ($L\_{H\_s}$).
> >
> >  More specifically, on the one hand, in **Theorem 2**, clipping weight controls the F-norm of the implicit gradient  $([\Delta W\_t]\_1^L/[G_t]\_1^L)$), which reduces the expected generalization error, on the other hand, we can evaluate the empirical risk ($L\_{H\_s}$) by observing the model's performance on the validation set. If the generalization error is determined, it is at least possible to estimate the population risk ($L_{\mu}$). Thus, the most challenging aspect is addressing the generalization error. For the direct theoretical analysis of empirical risk, we will consider this aspect in our future work.
>
> > **A to Q2** (Could this mean that fewer context examples are more robust for SVD weight pruning?):
> >
> >That's a pretty worthy question! Yes, fewer context examples can indeed lead to more robust results in SVD weight pruning.
> >
> > (1) Experimentally, in **Section 2.2**, we analyze the effect of different ICL shot numbers. As shown in Figure 2, with a decrease in the number of shots, the rate of performance collapse in the model slows down after a sharp reduction at the shallow layer.
> >
> > (2) Theoretically, $\Delta {W}(N)-\Delta {W}(N^{'}) = \left( \sum\_{i=N^{'}+1}^N{W}\_V{{h}\_i} \otimes {W}\_K{{h}\_i} \right) {W}\_Q,$ so it is supposed that the implicit gradient is more sensitive for SVD weight pruning.
> >
> > (3) The robustness mainly concerns the operation "SVD weight pruning", rather than the model's performance.
>
> > **A to Q3** (What would be the effect of pruning only a single module?):
> >
> > Good question! In our theory, the example (pruning only K or V) is provided in **Appendix B.6**, demonstrating how weight pruning can affect the norm of  $[\Delta W_t]_1^L$/$[G_t]_t^L$. Thus, it may confer advantages on the performance of Transformers in ICL inference. However, in **Theorem 2** (Remark 5), it also suggests that modifications to the shallow layers have a less steady impact.
> >
> > Additionally, please review the supplementary experimental results provided below. (We choose key projection matrix and select the  3-layer (large matrix condition number) of GPT-J-6B, others are the same to **Section 2.1**).
> >
> > | Task | Optimal $\xi$ | Accuracy/F1 Improve      |
> > | --------- | --------------------- | ------------------------ |
> > | SST-2     | 0.0                  | 0.7828 - 0. 7828 |
> > | RTE       | 0.5                   | 0.5413 - 0.5413   |
> > | COPA      | 0.995                 | 0.53 - 0.54 |
>
> > **A to Q4** (What would happen if we apply the same clipping rate to other datasets?):
> >
> > As we mentioned in **Section 3.3 and Section C.4**,  the implicit gradients produced by Transformers in practical applications are noisy due to factors such as the extent of model pre-training and data characteristics (eg. ICL shot number/task difficulty). Therefore,  $[\Delta W_t]_1^L$/$[G_t]_t^L$ in **Theorem 1** exhibit varying levels of noise, causing the optimal clipping rate to vary among different tasks, as it is dependent on the specific task and data. So if we apply the clipping rate (0.95) as used in SST-2 to other datasets, the model performance can either improve or deteriorate. Additionally, It is possible to conduct a certain number of experiments to find a range of optimal clipping rate that is broadly applicable.
>
> > **A to Limitations:**
> >
> > - For the equation between line 228 and 229.
> >
> > $||W^k\_{V\_r}h\_i^{k−1}+\delta\_Vh\_i^{k−1}||\_2^2=||W^k\_{V\_r}h\_i^{k−1}||\_2^2+2(W^k\_{V\_r}h\_i^{k−1})^T(\delta\_Vh\_i^{k−1})+||\delta\_Vh\_i^{k−1}||\_2^2=||W^k\_{V\_r}h\_i^{k−1}||\_2^2+||\delta\_Vh\_i^{k−1}||\_2^2$.
> >
> > Note that  $W^k_{V_r}=U_{:r}\Sigma_{:r}V_{:r}^T$ and $\delta_V=U_{r:}\Sigma_{r:}V_{r:}^T$, and $U_{:r},U_{r:}$ are **orthometric** (properties of SVD).
> >
> > Therefore, $(W^k_{V_r}h_i^{k−1})^T(\delta_Vh_i^{k−1})=(h_i^{k−1})^TV_{:r}\Sigma_{:r}(U_{:r}^TU_{r:})\Sigma_{r:}V_{r:}^Th_i^{k−1}=0$.
> >
> > - For the equation for about F-norm in lines 230,231,232. Thank you for pointing out the typo (231). We appreciate your attention to detail and will have made the necessary corrections. However, this minor typo does not affect the final conclusion. A detailed analysis is as follows:
> >
> >   For any vector $\mathbf{a}$ and $\mathbf{b}$, $\text{rank}(\mathbf{a}\otimes\mathbf{b})=1$, so the matrix ($\mathbf{a}\otimes\mathbf{b}$) only has **one** non-zero singular value.
> >
> >   Combined with $||A||\_F=\sqrt{\sum\_{i}\sigma\_i^2(A)}$ and singular value is nonnegative, we can get
> >
> >   $||\mathbf{a}\otimes\mathbf{b}||\_F = \sqrt{\sum\_{i}\sigma\_i^2(\mathbf{a}\otimes\mathbf{b})}=\max\_i[\sigma\_i(\mathbf{a}\otimes\mathbf{b})]$.
> > Therefore, the unique non-zero singular value will decrease after performing SVD on $\mathbf{a}$ and $\mathbf{b}$.

---

> > ### Comment · Reviewer_Zh6a · 2024-08-08
> > **Official Comment by Reviewer Zh6a**
> >
> > I appreciate the author's efforts to address my concerns, and I apologize for misunderstandings on my part. Upon further reflection, I find the theoretical analyses both reasonable and insightful in explaining the experimental results. While I still consider the use of 'gradient quality derived from context' (i.e., $G_t$) somewhat unconventional, I believe the theoretical contributions of this work are enough to explain the main observations presented in the paper. After reevaluating this study, I have decided to adjust the score I previously assigned.

---

> > > ### Author Response · Authors · 2024-08-09
> > > **Thank you**
> > >
> > > > Dear Reviewer Zh6a,
> > > >
> > > > Thank you for your thoughtful and constructive feedback regarding our manuscript. We deeply appreciate the time you invested in reevaluating our theoretical analyses and for acknowledging their relevance and insight in explaining the experimental results. Your meticulous review has truly enhanced the value of our work. Additionally, regarding the use of 'gradient quality derived from context' (i.e., $G_t$), please allow us to provide some additional clarifications (We will add a detailed description of this in the next version):
> > > >
> > > > - (1) In **Theorem 1** (Remark 2), $G_t$ is only dependent on $W_{t−1}$ and $H_s$, this is consistent with gradient descent in terms of relevance (Conventionally, gradients in training are only related to the current parameters and the training samples).
> > > >
> > > > - (2) In real-world training scenarios, SGD computes gradient by selecting a small batch of samples per iteration. This approach approximates the true gradient, inherently introducing noise. Similarly, in In-Context Learning, a small subset of samples (context examples) is used to generate implicit gradient (i.e., $G_t$), which also results in the introduction of noise.
> > > >
> > > > Once again, we thank you for your time and the constructive comments that have greatly enriched our work.

---

### Official Review · Reviewer_itEx · 2024-07-12

**Soundness:** 3
**Presentation:** 2
**Contribution:** 3
**Rating:** 5
**Confidence:** 3

**Summary:**

This paper discusses the phenomenon: SVD-based weight pruning can increase the in-context learning abilities of transformer based LLMs. In this paper, the authors conduct theorectical analysis by presenting the implicit gradient descent trajectories of ICL and providing the generation bounds visa full implicit gradient descent trajectories. This paper also provide a simple yet effective algorithm to clip the LLM by SVD to enhance ICL inference.

**Strengths:**

First, this paper has a clear writing and is easy to follow.

It provides a detailed theoretical analysis on why SVD based weight pruning will improve ICL performance by leveraging the implicit gradient descent trajectories. It also provides the generalization bounds of ICL, in Theorem 2, it can be inferred that the noise level and the norm of of gradient contribute to the error bound. It provides the theoretical insight of SVD based method.

The authors provides a simple algorithm to leverage the discovered phenomenon to improve ICL performance of LLM in a gradient-freee way. The ratio between $\sigma_{max} $ and $\sigma_{min}$ is a good choice of heuristic conditional number.

**Weaknesses:**

1. More details of algorithms is not shared. e.g. the range / number of clipping rate candidates set.
2. In experiments result of C.5, the optimal $\xi$ varies a lot across different tasks and different modules. However, this phenomenon is not touched in the theoretical part.

**Questions:**

1. Matrix condition number is an option for the indicator. But could there be more options, such as compute the decreasing rate of eigenvalues? Because when p=2, conditional number only leverages two values among all the eigenvalues.
2. Could authors provide further more clarification why optimal $\xi$ varies, and is there a way to explain this phenomenon under current theoretical framework provided in this paper?

---

> ### Author Rebuttal · Authors · 2024-08-02
>
> Thank you for your detailed review and valuable comments. Below we address specific questions.
>
> > **W1:** More details of algorithms is not shared. e.g. the range / number of clipping rate candidates set.
>
> > **A:** Firstly, the details of the algorithm can be reviewed in the **code** provided. Specifically, the set of clipping rate candidates is predefined. In our study, the clipping rate candidates are set as shown in **Figures 1 and 2**: [0, 0.1, 0.5, 0.75, 0.9, 0.95, 0.99, 0.995].
> > Besides, we analyze the impact of different hyperparameters through comparative experiments (details in **Section 2** and **Section 3.4**), as detailed below.
> >
> > - Clipping rate $\xi$：We search for the optimal $\xi$ in the predefined clipping rate candidates.
> >
> > - Predefined module $\mathcal{O}$ : The module containing the target pruning weights, which can be chosen from
> >   ['k_proj', 'q_proj', 'v_proj', 'out_proj', 'fc_in', 'fc_out', 'all', 'mlp', 'attn']
> >
> > - Selected layer $l$ : The layer containing the target pruning weights. For examlpe: In **Section 2**, we mainly focuse on comparing the impact of weight pruning to the **first two and the last two layers** of the model.
> >
> > - ICL shot number : The demonstration number in ICL, we analyze the effect of different ICL shot numbers in **Section 2.2**.
>
> ---
>
> > **W2:** In experiments result of C.5, the optimal $\xi$ varies a lot across different tasks and different modules. However, this phenomenon is not touched in the theoretical part.
>
> > **A:** That's a great question! The impact of different tasks and various module prunings on performance is indeed significant.
> >
> > However, our aim here is to provide a general analytical framework, so we have not delineated the impact of different layers/modules on performance in detail.
> >
> > In our theory, the simplification of the MLP and the handling method for the ATTN layer （mentioned in the main text） ：
> >
> > -  Feed-forward (MLP) layer：In **Appendix B.3**, it can be seen as a dimensional adaptation to correspond to residual connection. Then we can get the Implicit Gradient Descent with MLP:  $\Delta W_{icl}^{''}=W_{MLP}\left( \sum_{i=1}^N W_V{h_i} \otimes W_K{h_i} \right) W_Q$
> >
> > - Attention (ATTN) layer:
> >
> >   (i) In main text **Section 3.2**, our theoretical analysis is mainly focuses on linear attention setting.
> >
> >   $\hat{h}\_{N+1} = h\_{N+1}+ \Delta W\_{icl} h\_{N+1}, \Delta W\_{icl}=\left( \sum\_{i=1}^N W_V{h_i} \otimes W\_K{h\_i} \right) W\_Q$
> >
> >   (ii) In Section **Appendix B.2**,  it can be considered that the mapping function $\phi$, or rather the effect of the **Softmax** function, is to project the original features into a higher-dimensional space to capture more profound features. Subsequently, learning and meta-optimization are conducted under this **new feature space**.
> >
> >   $\hat{h}\_{N+1}=h\_{N+1} + \Delta W_{icl}^{'}\phi(W\_Q h\_{N+1}), \Delta W_{icl}^{'}=\frac{1}{D^{'}} \left[\sum_{i=1}^N (W_VH_s)_i \otimes \phi(W_KH_s)_i \right]$
> >
> > Then if one consider more than one layer,  a more detailed analysis can be provided within the current framework to depict the interaction relationship between the input feature and these different  modules.
> ---
> > **Q1:** Matrix condition number is an option for the indicator. But could there be more options, such as compute the decreasing rate of eigenvalues? Because when p=2, conditional number only leverages two values among all the eigenvalues.
>
> > **A:**  Good question! Let's first review our theory in **Theorem 2**: Modulating the norm of $[G\_t]_1^L$ or $[\Delta W\_{t}]_1^L$  could enhance performance when utilizing ICL.
> >
> > Our primary objective in this work is to provide a general analytical framework. The simple algorithm is employed to demonstrate the effectiveness of theoretical analysis in guiding experimental procedures. So, Any indicator that can guide the control of norms is a potential option. For instance:
> >
> > (i) Consider the behavior of the first few and last few singular values.
> >
> > (ii) Assessing the intrinsic rank of the matrix can be valuable, as it offers another dimension of understanding beyond just the condition number. [1] [2]
> >
> > [1] Armen Aghajanyan et al. Intrinsic Dimensionality Explains the Effectiveness of Language Model Fine-Tuning. URL https://arxiv.org/abs/2012.13255.
> >
> > [2] Edward Hu et al. Lora: Low-rank adaptation of large language models. ICLR 2022.
>
> ---
>
> > **Q2:** Could authors provide further more clarification why optimal $\xi$ varies, and is there a way to explain this phenomenon under current theoretical framework provided in this paper?
>
> > **A:** (i) As we mentioned in **Section 3.3 and Section C.4**,  the implicit gradients produced by Transformers in practical applications are noisy due to factors such as the extent of model pre-training and data characteristics (eg. ICL shot number/task difficulty). Therefore,  $[\Delta W_t]_1^L$/$[G_t]_t^L$ in **Theorem 1** have varies noise. That is why optimal $\xi$ varies.
> >
> > (ii) Besides,  expected generalization error (**Theorem 2**) = population risk - empirical risk ($L_{\mu}$-$L_{H_s}$).  we search the optimal $\xi$ on the validation set and subsequently evaluate it on the test set (detailed in **Section 3.4**).
> >
> > On the one hand, in **Theorem 2**, clipping weight controls the F-norm of the implicit gradient ( $[\Delta W\_t]\_{1}^{L}/[G\_t]\_{1}^{L}$), which reduces the expected generalization error, on the other hand, we can evaluate the empirical risk ($L_{H_s}$) by observing the model's performance on the validation set and estimate the population risk ($L_{\mu}$). (Note that different types of tasks affect $L_{H_s}$ differently, and different modules have varying impacts on the expected generalization error. See **A** to **W2**.)

---

> > ### Comment · Reviewer_itEx · 2024-08-12
> > **Thank you for your response**
> >
> > I apprecitated the authors' responses have successfully addressed my previous questions. I will hold my positive rating on this paper.

---

> > > ### Author Response · Authors · 2024-08-12
> > > **Thank you**
> > >
> > > > Dear Reviewer itEx,
> > > >
> > > > Thank you very much for your positive feedback. We truly appreciate your acknowledgment that our responses have successfully addressed your previous questions. Your continued support and positive rating of the paper are immensely encouraging. Thank you once again for your thoughtful review and valuable insights.

---

### Author Rebuttal · Authors · 2024-08-04

We are grateful to all reviewers for their detailed and constructive feedback! We are encouraged to see that reviewers find:

> - **Reviewer itEx**: It provides a detailed theoretical analysis on why SVD based weight pruning will improve ICL performance...... It provides the theoretical insight of SVD based method. The authors provide a simple algorithm to **leverage the discovered phenomenon** to improve ICL performance of LLM in a gradient-freee way.
>
> - **Reviewer Zh6a**: This work conducts an in-depth analysis to explain the "stability" of transformer weight pruning across different layers. The framework is interesting and validated through experiments. Moreover, the **theoretical analysis can be applied to design new algorithms** like algorithm 1 in this paper.
>
> - **Reviewer Nxqr**: The Authors provide a theoretical analysis to explain their empirical findings..... Furthermore, they propose a simple, derivative-free algorithm for enhancing ICL performance in downstream tasks, **demonstrating the practical value of their theoretical insights**.
>
> - **Reviewer CqwV**: The method intuitively makes sense and is something which can be **conditionally tuned after training** based on specific tasks if a validation set is available.

We have addressed all the questions raised by the reviewers through detailed clarifications, providing separate responses to each reviewer. Additionally, we would like to address some common concerns in a consolidated global response.

> **(1) The main contribution of this work.**
>
> Our primary objective is to provide a general **theoretical framework** that reveals the underlying mechanism behind the phenomenon that SVD-based weight pruning can enhance ICL performance.  Based on our theoretical insights, we can design new algorithms to enhance ICL performance. Consequently, we did not compare our approach with other pruning methods. Algorithm 1 is presented solely to illustrate how theoretical analysis can guide experimental procedures effectively.
>
> Specifically, in **Theorem 2**: Modulating the F-norm of $[G\_t]\_1^L$ or $[\Delta W\_{t}]\_1^L$​  could enhance performance when utilizing ICL.  So, there may be other potential **weight-based** pruning methods. For instance: magnitude-based pruning [1], which can directly control the $||A||\_F=\sqrt{\sum\_i\sum\_ja\_{ij}^2}$.
>
> Of course, there are also some **layer-based** pruning methods, as mentioned in (**Reviewer Nxqr Q1**) and (**Reviewer CqwV W3**). We would like to clarify the drawbacks of the drop-layer method under our theoretical framework：
>
> As stated in **Theorem 1**,  L layers with implicit gradient descent trajectories $[\Delta W_t]_1^L$/$[G_t]_t^L$. Consequently, dropping an entire layer leads to two main issues:
>
> (a) Adjusting the weights for other downstream tasks becomes more challenging.
>
> (b) The implicit gradient update is reduced by one step, which perhaps results in a decline in model performance.
>
> [1] Wen, W et al. Learning structured sparsity in deep neural networks. NeurIPS 2016.

> **(2) The explanation for simplifications.**
>
> To facilitate qualitative analysis, our main text primarily focuses on the theoretical aspects of linear attention, a considerable portion of the works also consider from the linear attention to explore the ICL [2,3].  And our goal is to provide insightful conclusions under simplified conditions.
>
> However, we also discuss the standard Softmax attention setting and MLP in **Appendix B.2** and **Appendix B.3**:
>
> Specifically, it can be considered that the mapping function $\phi$, or rather the effect of the **Softmax** function, is to project the original features into a higher-dimensional space to capture more profound features. Subsequently, learning and meta-optimization are conducted under this **new feature space**.
>
> $\hat{h}\_{N+1}=h\_{N+1} + \Delta W\_{icl}^{'}\phi(W\_Q h\_{N+1}),\ \Delta W\_{icl}^{'}=\frac{1}{D^{'}} \left[\sum\_{i=1}^N (W\_VH\_s)\_i \otimes \phi(W\_KH\_s)\_i \right]$
>
> For the MLP layer, it can be seen as a dimensional adaptation to correspond to the residual connection. Then, we can get the implicit gradient descent with MLP:  $\Delta W_{icl}^{''}=W_{MLP}\left( \sum_{i=1}^{N} W_V{h_i} \otimes W_K{h_i} \right) W_Q$
>
> [2] Ekin Akyiurek et al. What learning algorithm is in-context learning? investigations with linear models. ICLR 2023.
>
> [3] Johannes von Oswald et al. Transformers learn in-context by gradient descent. ICML 2023.

> **(3)  Why optimal clipping rate $\xi$ varies? How should Theorem 2 be interpreted?**
>
> (a) As we mentioned in **Section 3.3 and Section C.4** (which follows [4]),  the implicit gradients produced by Transformers in practical applications are noisy due to factors such as the extent of model pre-training and data characteristics (eg. ICL shot number/task difficulty). Therefore,  $[\Delta W_t]_1^L$/$[G_t]_t^L$ in **Theorem 1** have varies noise. **That is why** optimal $\xi$ varies.
>
> (b) Expected generalization error (**Theorem 2**) = population risk ($L\_{\mu}$) - empirical risk ($L\_{H_s}$).
>
> More specifically, on the one hand, **Theorem 2** shows that clipping weights controls the F-norm of the implicit gradient $([\Delta W\_t]_1^L/[G\_t]\_1^L)$ which helps reduce the expected generalization error. On the other hand, we can evaluate the empirical risk $(L\_{H\_s})$ by assessing the model's performance on the validation set. If the generalization error is known, it is possible to estimate the population risk ($L\_{\mu}$). Therefore, the most challenging aspect is addressing the generalization error.
>
> [4] Shivam Garg et al. What can transformers learn in-context? a case study of simple function classes. NeurIPS 2022.

Thank you once again to all the reviewers for your patience and invaluable comments. We hope our responses have clarified your initial concerns and questions. We are happy to provide further clarifications if necessary.

---

### Decision · Program_Chairs · 2024-09-25

**Decision:**

Accept (poster)

**Comment:**

The reviewers generally acknowledge the importance of research on in-context learning (ICL) and largely agree that the proposed method is intuitive and simple and that the manuscript is insightful. It was initially brought up, that the theoretical analysis was limited in that it only explains why stable performance is achievable, but not why better performance can be achieved with SVD-based weight pruning. After clarification in the discussion phase, reviewers appeared to be convinced that the presented analysis was sufficiently detailed and provides an explanation of the empirical results. The authors’ response to the question by reviewer CqwV, on whether dropping a layer entirely may sometimes be the best option compared to pruning a layer’s weights, seems reasonable.
Overall, all reviewers agree that the work meets the standards of a NeurIPS publication and I recommend acceptance.